# Systematic analysis of immune cell motility leveraging the open intravital microscopy database Immunemap

Diego Ulisse Pizzagalli [ID] [1,2,3,37 ✉], Pau Carrillo-Barberà [ID] [1,4,5,37], Himanshu Bansal [ID] [1,3], Elisa Palladino[1], Kevin Ceni[1,6], Benedikt Thelen[2], Alain Pulfer[1,7], Enrico Moscatello[2], Raffaella Fiamma Cabini [ID] [2], Johannes Textor[8,9], Inge M N Wortel[8,9], Immunemap Consortium*, Rolf Krause[2] & Santiago Fernandez Gonzalez [ID] [1,3 ✉]

## Abstract

**Understanding the spatiotemporal dynamics of immune cells in living organisms is a major goal in bioimaging. Intravital microscopy enables direct observation of cellular behavior over time with tissue-to-subcellular resolution, making it essential for investigating immune responses across tissues, conditions, and disease contexts. However, most intravital microscopy data remain siloed in individual labs, limiting reuse, standardization, and large-scale analysis. To address these limitations, we present Immunemap, an open-data platform and interactive atlas of immune cell motility. Immunemap currently provides access to over 58,000 curated single-cell tracks and more than 1,049,000 cell-centroid annotations from 400 intravital microscopy videos in murine models, spanning diverse tissues and conditions. The platform supports both exploratory and quantitative research. We show here how unsupervised learning identifies distinct motility patterns, and how large-scale mapping enables comparisons across stimuli, imaging setups, and organs. Its cloud-based architecture offers an interactive web interface and public APIs for integration with computational pipelines. By adhering to FAIR principles (Findability, Accessibility, Interoperability, and Reuse) and fostering cross-disciplinary studies, Immunemap supports reproducible research and provides a benchmark for bioimage analysis and tool development in intravital imaging.**

**Keywords** Intravital Microscopy; Cell Tracking; Spatiotemporal Dynamics; Immune Response; Bioimage Data Analysis
**Subject Categories** Computational Biology; Immunology; Methods & Resources

See also: M PeiselerI

## Introduction

The immune response encompasses complex biological processes with a dynamic component, such as cell recruitment, migration, and cell-to-cell interaction (Kurosaki et al, 2012; Crainiciuc et al, 2022). Live imaging is critical to understanding cell behavior in physiological and pathological conditions. Intravital microscopy (IVM) is particularly effective for investigating immune cell dynamics, allowing the observation of biological phenomena within living animals (Fig. 1A), typically in murine models, at a cellular resolution (Pittet et al, 2018). This method employs a range of microscopy techniques (e.g., multiphoton and confocal spinning disk), and different surgical models, allowing the proper exposure and immobilization of the studied organ. Indeed, in the last two decades, IVM has unveiled unprecedented mechanisms associated with the immune response, including antigen capture and presentation, tumor immune surveillance, elimination of infected or transformed cells, and the response to inflammatory stimuli, amongst many others (Pittet et al, 2018; Stein et al, 2017; Pizzagalli et al, 2022; Cahalan and Parker, 2008).

Typically, IVM protocols generate multidimensional data, up to 3D volumes acquired at multiple time points, including different channels (Fig. 1B, top). This data allows the simultaneous study of the spatiotemporal activity of multiple cell populations in situ. The conventional method for analyzing this data involves tracking individual cells and computing different motility parameters (Beltman et al, 2009) (Fig. 1B, bottom). Following these analyses, different movement patterns of immune cells have been described. Indeed, some of these patterns are associated with relevant biological functions such as cell recruitment (straight paths with high directionality and speed), immune surveillance (twisted path with low directionality resembling random movements), formation of immune synapses, and cell-mediated killing (often involving the arresting of the immune cell) (Pizzagalli et al, 2022). Hence, cell tracks and motility patterns represent a valuable source of information for understanding immune cell dynamics.

[1]Institute for Research in Biomedicine, USI, Bellinzona, Switzerland. [2]Euler Institute, USI, Lugano, Switzerland. [3]Faculty of Biomedical Sciences, USI, Lugano, Switzerland. [4]Departamento de Biologia Celular, Biologia Funcional y Antropologia Fisica, Universitat de Valencia, Valencia, Spain. [5]Instituto de Biotecnologia y Biomedicina (BioTecMed), Universitat de Valencia, Valencia, Spain. [6]Institute for Diagnostic and Interventional Neuroradiology, Inselspital, Bern University Hospital, Bern, Switzerland. [7]Department of Information Technology and Electrical Engineering, ETH Zurich, Zurich, Switzerland. [8]Medical BioSciences, Radboud University, Nijmegen, Netherlands. [9]Data Science, Institute for Computing and Information Sciences, Radboud University, Nijmegen, Netherlands. [37]These authors contributed equally: Diego Ulisse Pizzagalli, Pau Carrillo-Barberà. *A list of authors and their affiliations appears at the end of the paper.✉E-mail: pizzad@usi.ch; santiago.gonzalez@irb.usi.ch

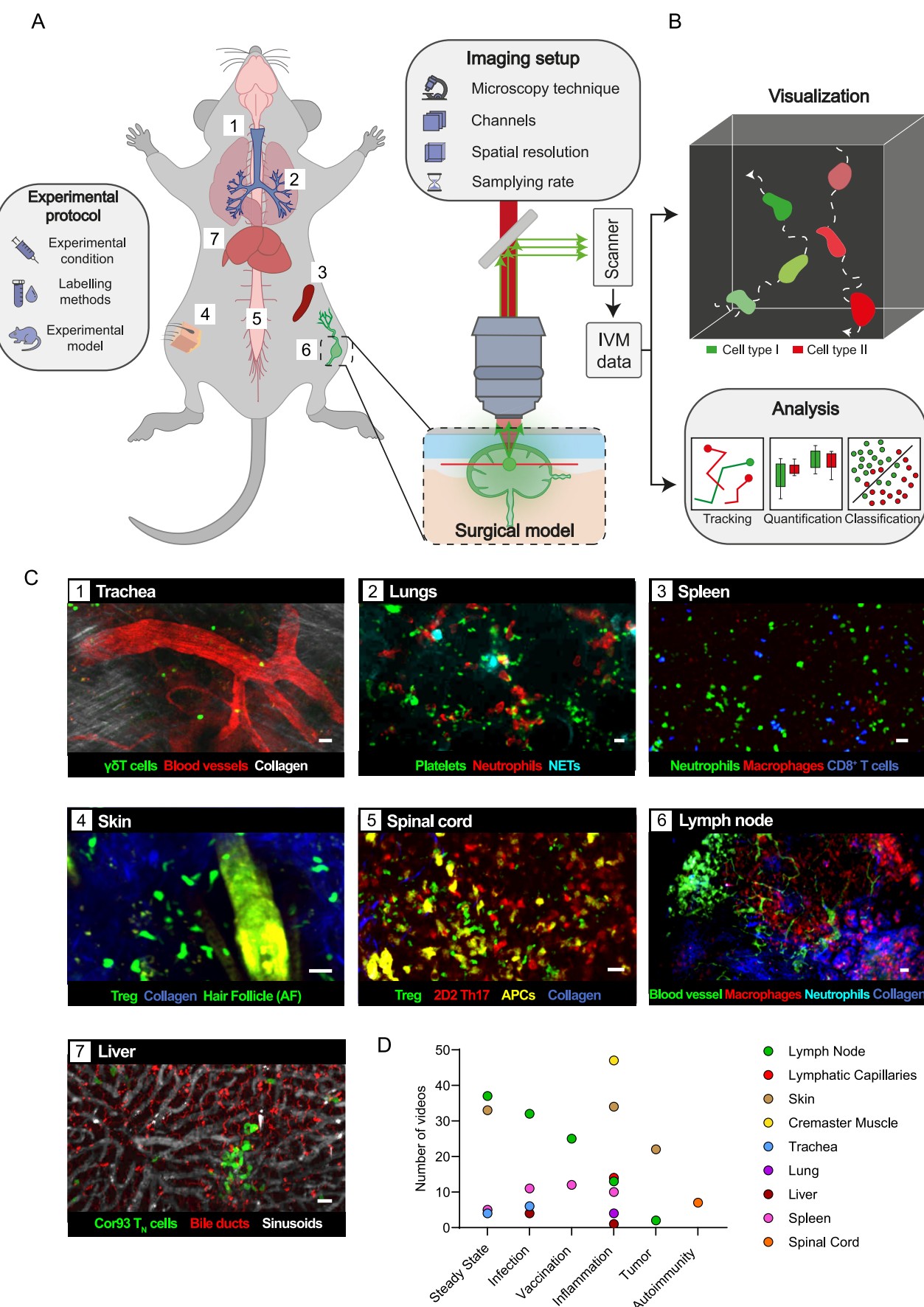

**Figure 1. Immunemap offers an open collection of intravital microscopy data from immunological studies, captured across various organs and experimental conditions.**

(A) A schematic overview of the intravital microscopy (IVM) imaging pipeline, using a representative surgical model—the murine popliteal lymph node. The experimental setup includes the selection of the cell labeling method, treatment, and appropriate surgical model. Fluorescently labeled cells are excited using a light source (e.g., multiphoton laser), and a scanner records the emitted fluorescence to generate digital imaging data. (B) Analysis of the multidimensional dataset generated via IVM, composed of 3D volumes acquired across successive time points and multiple spectral channels. This includes specialized software for visualization and quantification of cell behavior via tracking individual cells, extraction of motility-related features (e.g., speed, directionality, etc.), and subsequent classification based on motility parameters. (C) Depiction of representative IVM data available in Immunemap that were acquired in a variety of anatomical locations: trachea (1), lung (2), spleen (3), skin (4), spinal cord (5), lymph node (6), and liver (7). AF autofluorescence, APCs antigen-presenting cells, NETs neutrophil extracellular traps. Scale bar = 20 μm. (D) Distribution of the number of videos in Immunemap by anatomical location and experimental condition. Source data are available online for this figure.

However, only a small portion of the information from an IVM acquisition is made available to the research community alongside the published manuscript for further analysis. Raw data remains stored on private servers, restricting broader access for the scientific community. Typically, compressed or processed data is available as supplementary material, while raw data can only be obtained through an explicit request to the authors. Furthermore, metadata and experimental details are often not presented in a machine-readable format and can be sparse and fragmented within the manuscript. Lastly, only statistical analyses of motility parameters are included in the manuscript, while tracks of individual cells are generally not provided. Overall, this process lowers the reproducibility of the experiments, resulting in information loss and hindering other groups from reusing previously acquired IVM data for further analysis.

The availability of large, well-annotated datasets has been instrumental in the emergence of new research fields, such as bioinformatics (Garcia et al, 2019), deep learning (Asuncion and Newman, 2007), and in the development of advanced algorithms for analyzing complex datasets (Goldberger et al, 2000). In parallel, Open Data initiatives guided by the FAIR principles (Findability, Accessibility, Interoperability, and Reusability) are becoming increasingly prominent in promoting the sharing of scientific data in a transparent and efficient manner. Resources such as the ImageDataResource (Williams et al, 2017), CellImageLibrary (Orloff et al, 2012), and software solutions like the OMERO server (Allan et al, 2012) have expanded access to imaging datasets. However, these platforms are not specifically designed to accommodate IVM data and do not provide the extended, dynamic datasets required to study immune cell behavior in vivo.

To address this gap, we introduce Immunemap (www.immunemap.org), a dedicated atlas of immune cell motility in living tissues. Built on an open data infrastructure aligned with FAIR principles, Immunemap enables efficient access, integration and reuse of IVM data at scale. With a growing repository of high-quality videos and expertly curated single-cell tracks, Immunemap offers a unique and comprehensive resource for the immunology and imaging communities. This platform not only facilitates reproducibility and transparency but also opens new avenues for data-driven discovery in immune cell dynamics.

## Results

### A living map of immune cell motility across tissues

An international consortium of 30 leading laboratories with recognized expertise in IVM has come together to build the Immunemap collection of IVM videos. Together, they have contributed over 400 videos captured across diverse anatomical sites including the trachea, lung, spleen, skin, spinal cord, lymph node, and liver, and both lymphatic and blood vessels providing a broad representation of immune cell behavior in vivo (Fig. 1C). Additionally, the videos were acquired under a range of experimental conditions, including steady-state, infection, vaccination, inflammation, tumor, and autoimmunity (Fig. 1D).

To better understand the scope and characteristics of the dataset, we analyzed the physical and technical properties of the IVM videos included in Immunemap. On average, videos had a mean time interval of 34 s between consecutive frames (Fig. 2A), an average duration of 32 min (Fig. 2B), and comprised ~61 frames per video (Fig. 2C). The typical field-of-view had a pixel size of 0.7 μm (Fig. 2D), a diagonal of 250 μm (Fig. 2E) and an average depth of 61 μm (Fig. 2F).

We also conducted a quality assessment based on five key parameters that address common issues in IVM: signal variation over time (SV), photobleaching (PB), saturation (SAT), contrast (CR), and noise ratio (NR) (Fig. 2G). These parameters were combined into a single composite score, normalized between 0 and 1, to represent the overall quality of each video. The resulting distribution showed a near-normal trend centered around a mean score of 0.68 (Fig. 2H), with deviations primarily reflecting imaging artifacts encountered during acquisition, such as tissue drifting (Fig. 2I,J), compared to a high-quality video (Fig. 2K).

Building on this dataset, we next focused on generating single-cell tracks that allow the quantification of immune cell dynamics across conditions and tissues. Cell tracking is a fundamental step for extracting motility parameters and understanding behavior over time. To achieve broad representation across immune cell types (Pizzagalli et al, 2019b; Steinman, 2007; Palomino-Segura et al, 2023; Wilson et al, 2016), we performed 2D manual tracking on 233 videos that met predefined quality criteria, allowing reliable identification of individual cells. A team of five trained operators carried out the annotations using TrackMate software (Ershov et al, 2022; Tinevez et al, 2017), following standardized guidelines to ensure consistency and reproducibility (Appendix Fig. S1A).

This effort resulted in a comprehensive collection of 58,943 manually annotated 2D cell tracks, accounting for 1049340 centroid annotations. Amongst these, 2295 tracks corresponded to B cells; 25573 to T cells—including 10347 CD4+, 13,056 CD8+, 782 γδ T cells and 1388T cells of other subtypes. 6244 NK cells; 11,047 neutrophils; 894 eosinophils; 1952 to innate lymphoid cells; and 241 to dendritic cells. The remaining tracks included other cell types that interact with the host immune system, such as platelets and humanized cells (Fig. 3A). On average, individual tracks

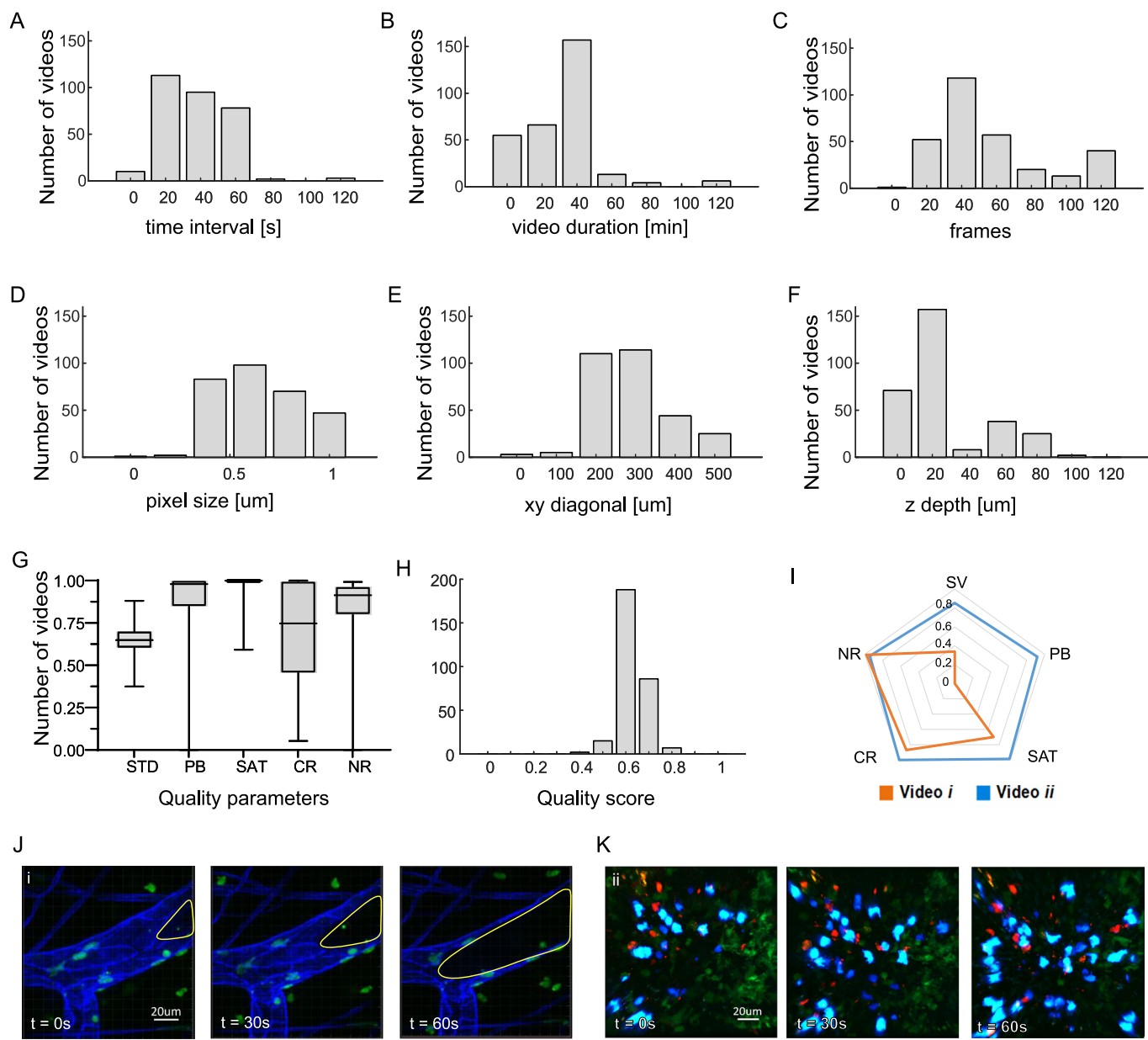

**Figure 2. Properties of the imaging dataset included in Immunemap.**

(A) Time interval between consecutive frames. (B) Total video duration (in minutes). (C) Number of frames per video. (D) Pixel size in the xy-plane. (E) Diagonal length of the field-of-view (xy). (F) Imaging depth along the z-axis. (G) Distribution of individual image quality metrics, including signal variation over time (SV), photobleaching (PB), saturation (SAT), contrast (CR), and noise ratio (NR). Box plots show the median (black central line), interquartile range (box bounds), minima and maxima (whiskers). Data are from $n = 298$ videos. (H) Distribution of the composite quality score derived from the individual quality metrics shown in (G). (I) Example radar plot illustrating the individual quality parameters for two representative videos. (J, K) Representative video sequences highlighting the presence (J) or absence (K) of imaging artifacts. In (J), loss of fluorescence over time due to tissue drift is visible in a vessel (yellow line), while (K) shows a stable recording without major artifacts. Source data are available online for this figure.

spanned 8 min ($\pm$1 min), contained around 21 annotations, and covered an average displacement of 42 μm (Fig. 3B), enabling detailed visualization of cell behavior within their native micro-environments (Fig. 3C).

A descriptive analysis of motility parameters across all annotated tracks indicated distinct behavioral profiles among immune cell types. CD8$^+$ T cells exhibited the highest motility, with a median speed of 8.9 μm/min, directionality of 0.68, and an arrest coefficient of 0.05. In

contrast, dendritic cells were the least motile, with a median speed of 3.2 μm/min, directionality of 0.23, and an arrest coefficient of 0.41 (Fig. 3D–F). The large interquartile ranges further indicated the possibility of each cell type to undergo different migration patterns. This was made further evident from a bivariate analysis, showing a non-uniform distribution characterized by multiple peaks (Fig. 3G,H). Though this analysis provided an overview of these patterns, results might change as Immunemap keeps growing.

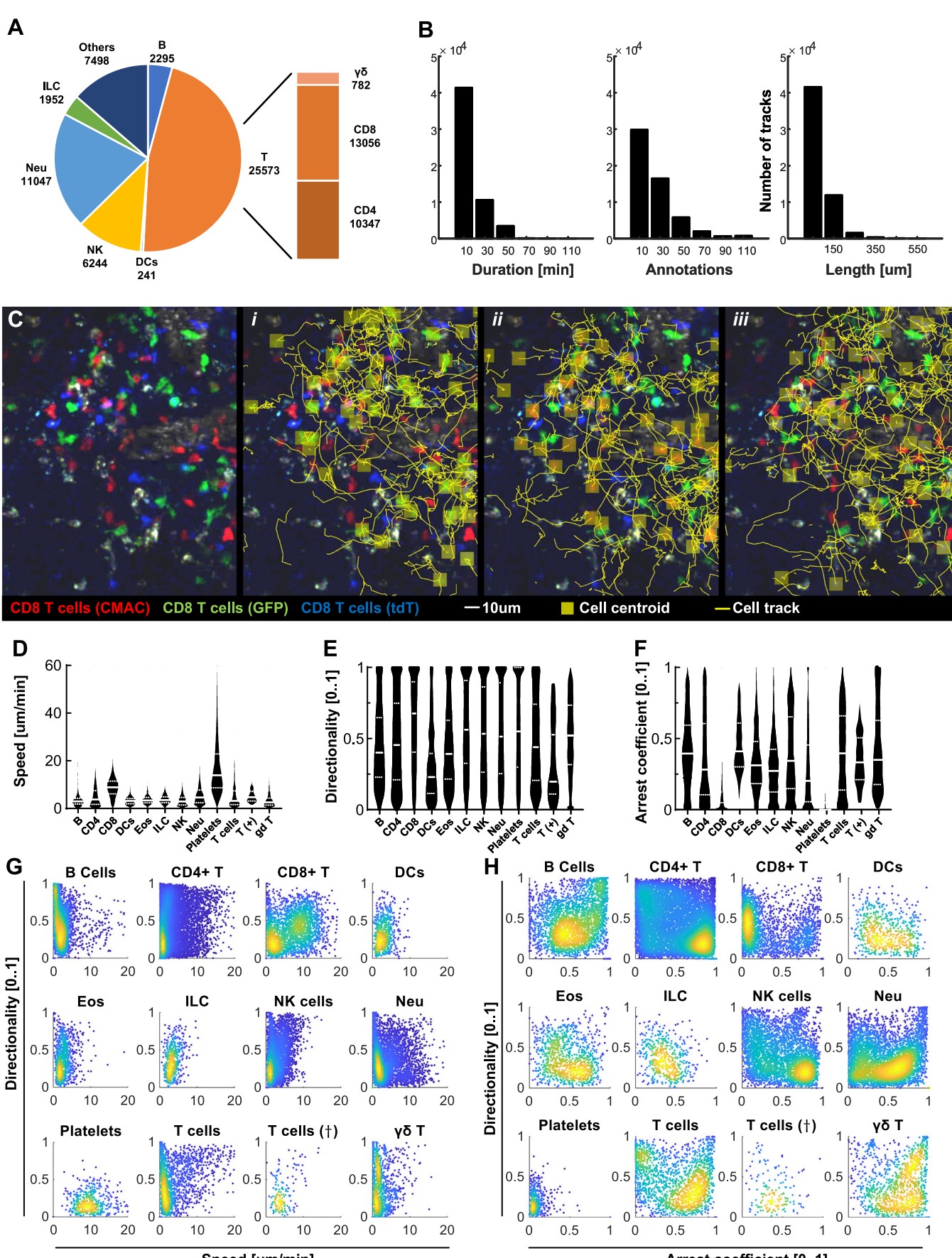

**Figure 3. Tracks of immune cells included in Immunemap.**

(A) Pie plot representing the number of tracks per cell type (DCs stands for dendritic cells; NK natural killer cells, Neu neutrophils, ILC innate lymphoid cells; Others include additional cell types such as platelets and T cells in humanized mouse models). (B) Distribution of track duration (left), number of manually annotated cell centroids per track (middle), and track length (right). (C) Representative micrographs of IVM videos included in Immunemap showing the tracks for three different cell types (yellow lines) (i): GFP-labeled, (ii): CMAC-labeled, (iii): tdT-labeled CD8 + T cells. (D–F) Violin plots representing the distribution of general motility parameters per cell type: average speed (D), directionality (E), and arrest coefficient (F). The continuous white line represents the median, and the dashed white line represents the interquartile ranges. Data from $n = 2295, 10,347, 13,056, 482, 894, 1952, 6244, 11,047, 7416, 1306, 82$, and 782 tracks in each group, respectively. (G, H) Scatter plots showing the distribution of motility parameters (directionality, speed, and arrest coefficient) across cell types. Source data are available online for this figure.

To extend the analytical versatility of the dataset, we also reconstructed 3D trajectories by mapping each track's x and y coordinates to the z-plane with maximum intensity (Appendix Fig. S1B). We validated this algorithm using a subset of videos previously annotated in full 3D (Pizzagalli et al, 2018), confirming that the estimation of the z value was very accurate, as the $R^2$ value was higher than 0.8 for most of the tested tracks (Appendix Fig. S2K).

While 2D tracking remains widely used for studying cell motility, this method can underestimate key parameters such as speed, path length and displacement, ignoring movement along the z axis. To better assess the impact of 2D vs. 3D tracking on the motility measurements, we compared the resulting motility metrics —speed, displacement, directionality and arrest coefficient— between the original 3D trajectories and their 2D-projected counterparts from the dataset mentioned above. Results confirmed that dimensional reduction can introduce discrepancies across some of the evaluated metrics, such as an underestimation of the cell displacement (Appendix Fig. S2A–C) and mean speed (Appendix Fig. S2D,E), as well as an overestimation of the arrest coefficient (Appendix Fig. S2F,G), while no significant changes were observed in other parameters such as the directionality (Appendix Fig. S2H,I). These results underscored the importance of dimensionality in accurate motility quantification.

## Data-driven profiling of in vivo immune cell motility patterns

To characterize the primary motility patterns exhibited by immune cells, we conducted a motility analysis of 14,877 individual cell tracks using unsupervised machine learning. These tracks were chosen by filtering based on track duration (≥500 s) to quantify cellular dynamics having adequate time periods, as previously described (Antonello et al, 2023; Pizzagalli et al, 2019b). To enhance comparability among tracks of varying sampling rates, we interpolated all tracks to a uniform sampling rate of 10 s. Next, we computed directionality, square displacement, mean speed, and arrest coefficient features over a 50-s sliding window (comprising five-time points of 10 s each). These multivariate features of all tracks were processed via the dynamic time warping (DTW) algorithm to calculate the similarity of tracks. To visualize the $N \times N$ distance matrix computed by DTW, we reduce it to two dimensions using UMAP as previously described (Dekkers et al, 2023). This will also help to get interpretable clustering results in the next step.

This procedure enabled the application of clustering to identify tracks with similar motility parameters (Fig. 4A). In contrast to previously reported unsupervised analyses of cell motility based on k-means clustering (Dekkers et al, 2023), we relied on an algorithm

based on density peaks and network theory that avoids artefacts in presence of clusters with non-convex shapes (Pizzagalli et al, 2019a).

We further characterized each of the ten identified clusters by computing the average motility metrics of all the tracks included (Fig. 4B), and by visual inspection of representative tracks (Fig. 4C). Among them, cluster 1 was characterized by high directionality, low arrest coefficient, and intermediate speed, indicative of cells undergoing directed migration. Visual inspection of representative tracks confirmed that cells in this cluster moved towards one direction for a certain amount of time. Cluster 10 was characterized by an opposite pattern, with low speed, low directionality, low displacement, and high arrest coefficient— features consistent with a largely arrested cell population. In addition, cluster 7 was characterized by an intermediate phenotype, combining a high arrest coefficient with moderately high directionality, suggesting alternating phases of directional movement and arrest. Instead, cluster 5 included tracks with high displacement and low directionality, consistent with long, meandering trajectories. Cluster 6 contained the tracks with the highest displacement and speed; however, this cluster included only four trajectories and likely reflects tracking artifacts rather than true biological behavior. The other clusters represented intermediate subtypes of patrolling with varying degrees of directionality (Fig. 4B).

To better understand the diversity of immune cell migration behaviors, we analyzed the relative distribution of the ten motility clusters across different immune cell types tracked in Immunemap (Fig. 4D). When examining cluster usage by cell type, neutrophils were predominantly found in clusters 8 and 10, consistent with a mix of patrolling and arrested behaviors. CD8+ T cells showed strong representation in clusters 8 and 3, suggesting a combination of persistent and moderately exploratory motility. CD4+ T cells were enriched in clusters 7 and 5, suggesting more variable motility, alternating between directional movement and meandering. Eosinophils also concentrated in cluster 8 but showed an unusually high proportion in cluster 1 compared to other cell types, indicating episodes of more directed movement. Dendritic cells showed a broad distribution, with substantial representation in clusters 2, 5, and 10, reflecting their heterogeneous motility states. B cells were mainly distributed across clusters 5, 8, and 3, while NK cells were more evenly divided between clusters 3, 5, and 7. Finally, γδ T cells showed a predominant presence in cluster 10 and a notable contribution to cluster 8, suggesting reduced motility in the studied contexts.

In summary, this data-driven analysis highlights a wide range of movement patterns exhibited by immune cells, associated with their versatility and ability to mediate various biological processes (Pittet et al, 2018; Pizzagalli et al, 2019b; Steinman, 2007;

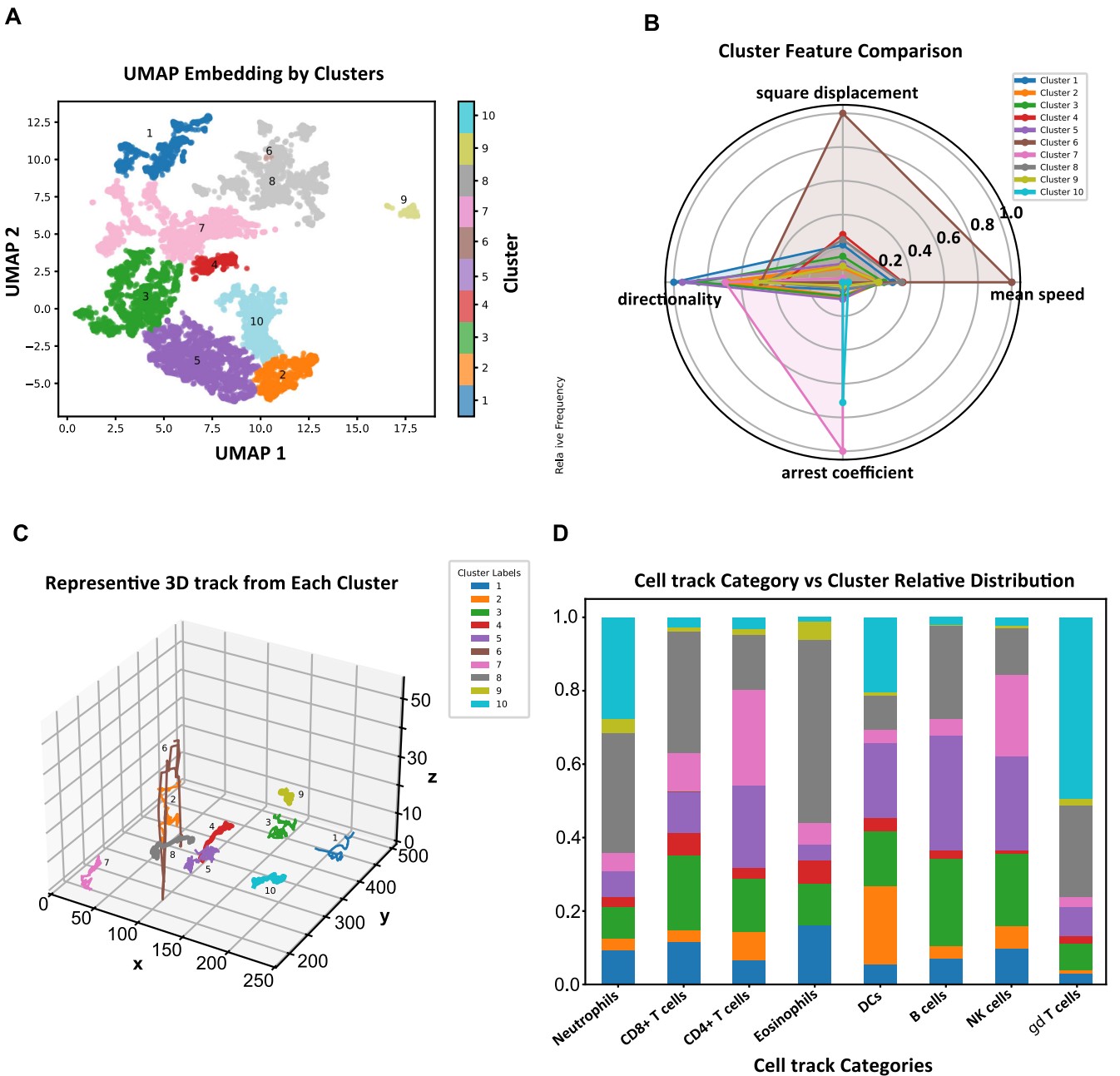

**Figure 4. Characterization of the main motility patterns displayed by immune cells via dynamic time warping and unsupervised learning.**

(**A**) Scatter plot showing the similarity in cell trajectories calculated by dynamic time warping (DTW) in a UMAP-reduced dimensionality space. Colors represent the ten clusters attributed via density peak clustering. (**B**) Radar plot showing motility properties associated with each cluster. Representative cluster features included: Cluster 1, consistent with directed migration; Cluster 10, largely arrested cells; Cluster 7, alternating phases of movement and arrest; Cluster 5, long meandering trajectories; and Cluster 6, likely tracking artefacts due to unusually high speed and displacement. The remaining clusters reflected variations of patrolling behavior with differing levels of directionality. Values indicate the average metric over all the tracks included in each cluster, normalized using the min-max normalization for radar chart interpretability. (**C**) 3D cell tracks representative of each cluster. (**D**) Distribution of the ten motility clusters among the different immune cell types tracked in Immunemap. Source data are available online for this figure.

Palomino-Segura et al, 2023; Wilson et al, 2016). While the number of annotated tracks per cell type remains limited and conclusions should be interpreted cautiously, this analysis illustrates the power of standardized, open-access resources to uncover emerging biological trends from aggregated datasets.

## Large-scale mapping of immune cell motility across diverse experimental conditions

By leveraging tracks and metadata provided by Immunemap, we first assessed how acquisition settings influence motility

## Effect of different acquisition settings cell measured motility metrics

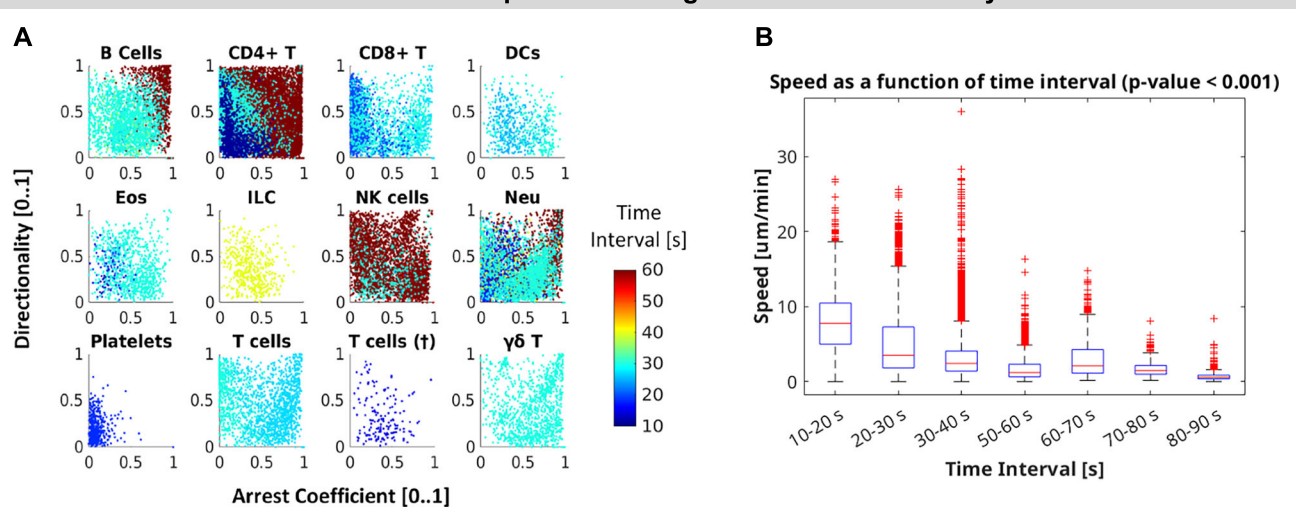

## Comparison of cell motility at steady state and upon inflammatory stimuli

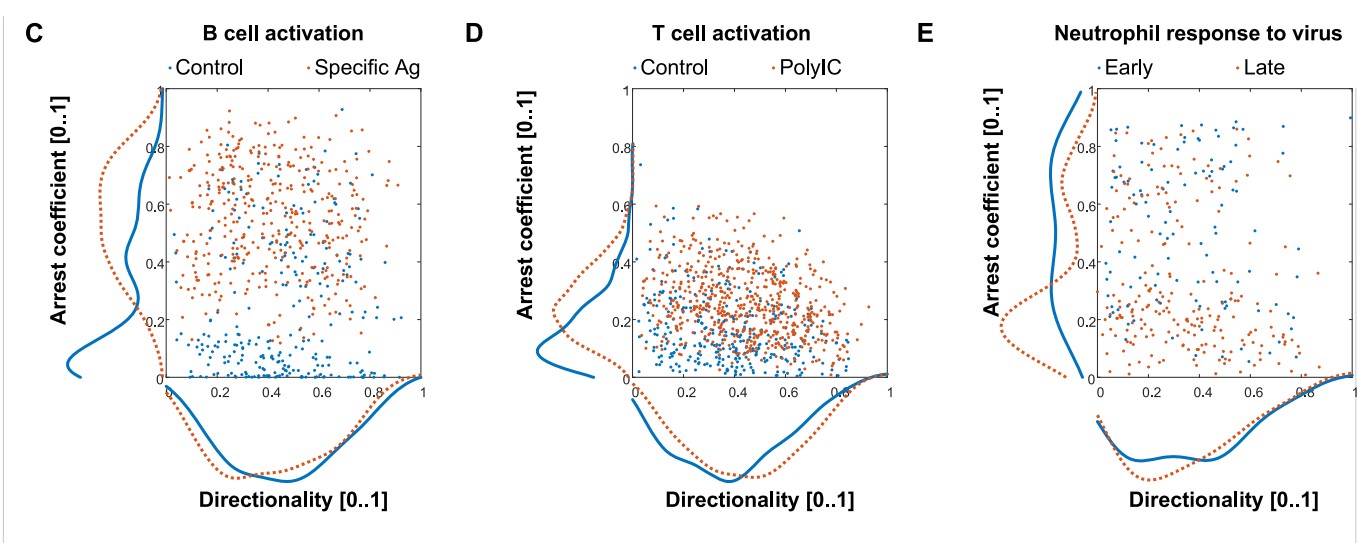

## Comparison of neutrophil motility under inflammatory conditions at different imaging sites

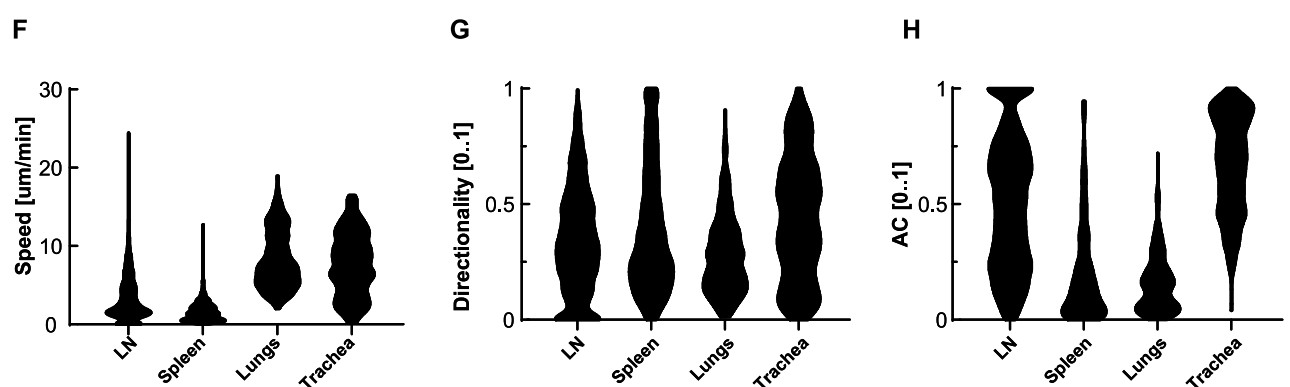

**Figure 5.  Immunemap enables large-scale comparative analysis of immune cell motility across diverse experimental conditions.**

By providing access to a curated and standardized repository of intravital imaging data, Immunemap allows researchers to perform quantitative, cross-condition analyses of immune cell behavior. The platform facilitates the evaluation of how cell motility is influenced by factors such as temporal resolution, inflammatory stimuli, and tissue-specific environments—supporting data-driven discovery and hypothesis generation in immune imaging. (A, B) Effect of temporal sampling rate on motility measurements. (A) Color-coded scatter plots showing the relationship between directionality and arrest coefficient across varying temporal resolutions. (B) Exponential decay in measured speed as a function of the time interval between consecutive image acquisitions. Data points represent 500 s track fragments from all Immunemap tracks with total durations >500 s. Box plots show the median (red central line), interquartile range (box bounds), minima and maxima (whiskers), and outliers (red cross). Statistical significance was determined using the Kruskal–Wallis test, exact $p$ value = $2.22 \times 10^{-16}$ (C–E) Immune cell responses to inflammatory stimuli. Blue dots represent tracks at steady state; orange dots indicate tracks of cells under inflammatory stimuli. Lines indicate the marginal distributions. (C) B cells under steady-state conditions (blue) versus after antigen-specific stimulation (orange). (D) OT-I CD8$^+$ T cells at steady state (blue) and following Poly I:C stimulation (orange). (E) Neutrophils at early (blue) and late (orange) time points after injection of UV-inactivated PR8 virus. (F–H) Tissue-specific differences in neutrophil motility. Neutrophils exhibit increased speed in the airways compared to lymphoid tissues, highlighting how the tissue microenvironment influences migratory behavior. Data from $n$ = 2515, 1621, 452, and 222 tracks in each group. Source data are available online for this figure.

measurements. We found that the temporal sampling rate ($\Delta t$) had a marked effect on all motility metrics. Directionality and arrest coefficient were tightly linked to the sampling interval, and an inverse relationship was observed between $\Delta t$ and measured speed, with longer intervals leading to an apparent reduction in velocity (Fig. 5A,B). These findings emphasize the need to account for acquisition parameters when comparing cell motility across different videos and highlight how Immunemap enables stratified analysis based on imaging settings.

Moreover, we compared in a selected number of samples immune cell motility at steady-state and under inflammatory conditions. Upon antigen-specific stimulation, B cells showed a shift associated with decreased speed and directionality and increased arrest coefficient with respect to the steady state (Fig. 5C). These parameters varied in a similar manner in OT-I CD8$^+$ T cells following Poly I:C treatment (Fig. 5D). In contrast, upon stimulation with an influenza vaccine, neutrophils increased their speed, as observed at later time points post-injection (Fig. 5E) with respect to earlier time-points. These results are consistent with known differences in how lymphocytes and neutrophils respond to activation and demonstrate the ability of Immunemap to capture such shifts across diverse experimental contexts.

As a final example, we analyzed neutrophil motility across different imaging sites, acquired under similar conditions (induction of inflammation). We found that neutrophils imaged in the airways (lung and trachea) displayed significantly higher speed compared to those in lymphoid tissues (lymph node and spleen), reflecting possible differences in the physiological conditions in these organs (Fig. 5F). However, the directionality and arrest coefficient were closer between spleen and lungs (Fig. 5G,H, respectively). Altogether, this analysis exemplifies how Immunemap supports large-scale, comparative evaluation of immune cell motility across different videos, and suggests which parameters should be considered when matching them.

## A cloud platform for sharing and analyzing immune cell imaging data

Understanding immune cell dynamics requires an interdisciplinary collaboration across immunology, imaging, and computational science. To support this, we developed Immunemap (www.immunemap.org), a cloud-based, open-data platform designed to centralize and disseminate intravital microscopy

datasets while guaranteeing long-term scalability through a distributed file system (Appendix Fig. S3A–B).

Such a platform was designed to host imaging data, metadata, experimental details, and single-cell tracks (Fig. 6A). To ensure cross-study comparability, each dataset entry has been manually curated. Metadata includes structured information about the imaging system, institution, microscope settings (e.g., pixel size, acquisition interval), staining protocols, imaging channels, mouse models, anatomical location, treatment conditions, and study references (Dataset EV1).

Immunemap offers a public web interface that allows users to browse, filter, and retrieve datasets (Fig. 6B, i). An integrated in-browser viewer enables rapid visualization of both videos and cell tracks, eliminating the need for downloads or third-party software (Fig. 6B, ii).

To support advanced computational analyses, Immunemap provides application programming interfaces (APIs) that deliver content in several formats, including videos in HDF5, metadata in JSON, and cell tracks in both CSV and JSON formats (Fig. 6B, iii). These APIs allow seamless integration with analysis pipelines in R, Python, or Matlab. For example, we implemented a module that imports Immunemap data directly into the CelltrackR (Wortel et al, 2021) package in R, facilitating the computation of motility metrics and statistical analysis. Using this pipeline, we analyzed the motility of CD8$^+$ T cells subjected to different nucleofection protocols and found that Thy1-targeted nucleofection did not significantly alter their migration behavior in lymph nodes (Appendix Fig. S4; Suppl. Material 1).

Additionally, we demonstrate how the datasets in Immunemap can support image-based methods beyond tracking. In videos with high cell density or low resolution, where tracking individual cells is challenging, we apply optical flow to estimate collective cell motility at the organ level (Appendix Fig. S5A–C). This technique generates flow fields (displacement of each pixel between adjacent frames), which can be visualized as motility heatmaps highlighting areas of increased or decreased activity in the organ (Pizzagalli et al, 2019b). Conversely, centroid annotations from Immunemap can be used to train deep learning models—such as encoder-decoder architectures—to automate cell detection in complex imaging conditions (Appendix Fig. S5D,E).

Data upload is enabled through a user dashboard (accessible upon registration), and Immunemap supports tracking outputs from commonly used software, including FIJI/TrackMate and

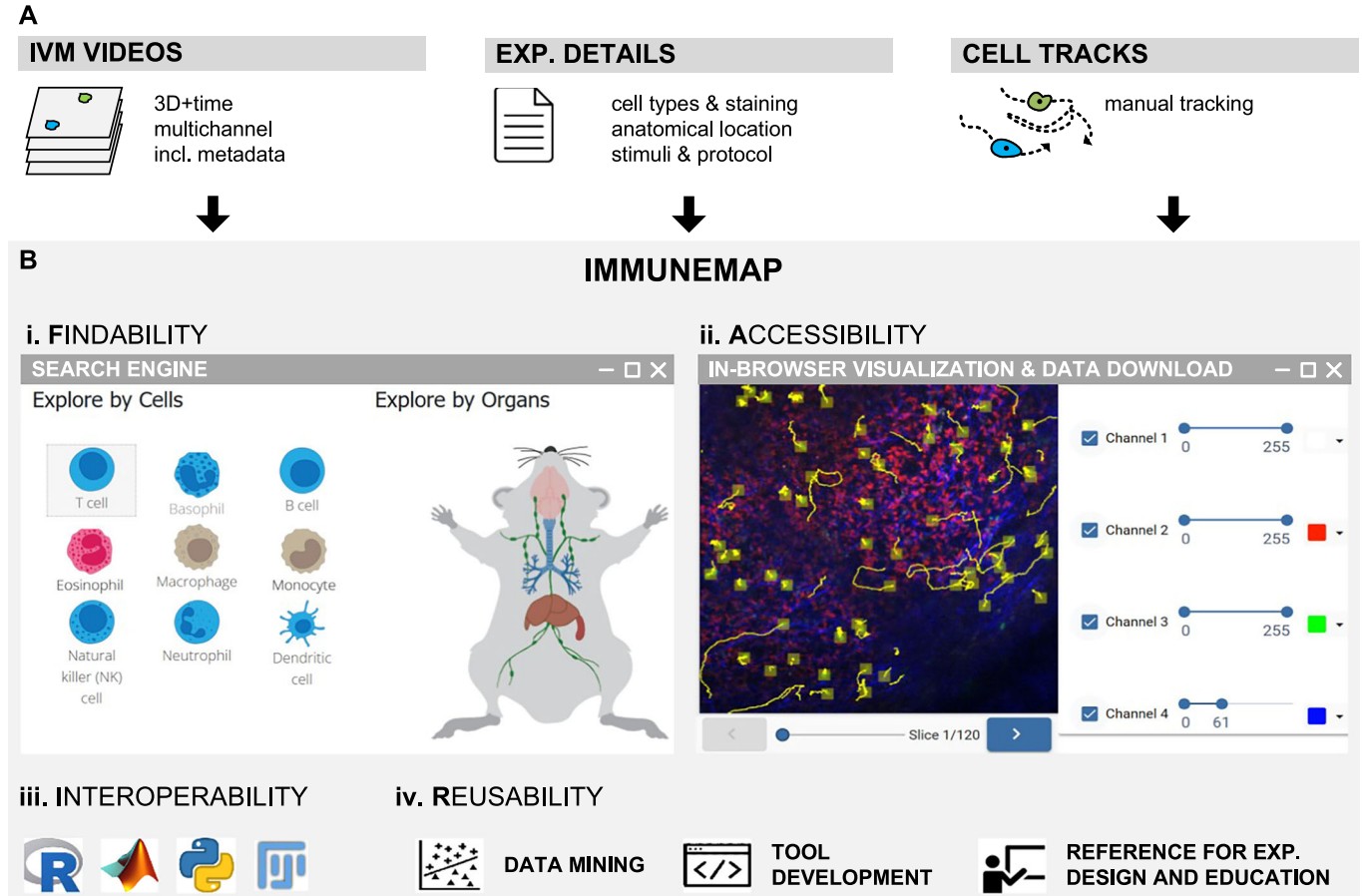

**Figure 6.  Immunemap promotes Open Research Data practices across research communities.**

(**A**) Overview of the Immunemap pipeline, which enables the curation, standardization, and centralization of intravital microscopy (IVM) data. This includes time-lapse multichannel 3D videos, detailed experimental metadata, and manually curated immune cell tracks. (**B**) Schematic representation of how Immunemap supports the FAIR principles (Findable, Accessible, Interoperable, and Reusable) for open data in bioimaging and immunology: (i) Findability—through a searchable online platform that allows users to browse and filter experiments by immune cell type, organ, or experimental condition. (ii) Accessibility—via in-browser visualization tools for videos and tracks, as well as direct download options for raw and processed data. (iii) Interoperability—through public APIs and ready-to-use scripts that allow seamless integration of Immunemap data into common analysis environments such as R (e.g., CelltrackR), Python, Matlab, and Fiji. (iv) Reusability—enabling downstream applications such as large-scale data mining, method benchmarking, algorithm development, experimental design guidance, and educational use.

Imaris. Once uploaded and integrated, datasets become publicly available and reusable—whether for extracting new biological insights, building and benchmarking analysis tools, or for use in teaching and training (Fig. 6B, iv).

Altogether, Immunemap provides a scalable, FAIR-compliant infrastructure that empowers the reuse of IVM data across disciplines, enhancing transparency, reproducibility, and discovery in immune cell imaging.

## Discussion

Studying cell motility is fundamental for understanding how immune cells interact with their microenvironment, respond to dynamic stimuli, and execute context-dependent functions. Cell tracking derived from time-lapse imaging provides a rich source of spatial and temporal data, enabling the quantification of complex behaviors. When paired with transcriptomic and spatial omics datasets, motility analysis contributes a dynamic, fourth dimension that complements molecular snapshots of cell state and location (Cui et al, 2024; Gorenshteyn et al, 2015). These multidimensional datasets are crucial for decoding cellular heterogeneity, evaluating the effects of pharmacological interventions (Beltman et al, 2009; Thelen and Stein, 2008; Friedl and Weigelin, 2008), and understanding emergent tissue-level phenomena.

In genomics and transcriptomics, public repositories coupled with machine learning techniques have revolutionized data reuse and discovery (Goldberger et al, 2000; Orloff et al, 2012; Bray et al, 2017; Clough and Barrett, 2016). A similar transformation is underway in bioimage analysis, where computer vision methods are being used to model cell morphology and behavior (Edlund et al, 2021). However, in IVM, these approaches remain underutilized, largely due to the absence of curated, large-scale, and accessible datasets. Previous initiatives have provided access to a limited number of cell tracks, typically aimed at benchmarking tracking algorithms rather than uncovering biological insights (Pizzagalli

et al, 2018; Ulman et al, 2017). In this context, Immunemap establishes the first open-data atlas of immune cell motility, offering not only a curated collection of microscopy videos and single-cell tracks, but also structured experimental metadata critical for reproducible analysis.

However, a challenge when integrating data from multiple sources is ensuring comparability across imaging conditions. Differences in acquisition parameters (such as field-of-view, pixel size, time interval, labeling protocols, or organ-specific setups) can introduce biases in motility metrics. Immunemap addresses this through rigorous manual curation and by associating each dataset with standardized metadata (Appendix Table S1), as well as automated quality assessments based on five imaging parameters (Fig. 2G–I). These data serve as a basis for normalization strategies, and tutorials are provided in Suppl. Material 1 demonstrates how metadata can be incorporated into statistical models to correct for acquisition bias (Ronteix et al, 2022). Moreover, Appendix Fig. S4 illustrates how variations in imaging setup can systematically affect derived motility metrics—underscoring the importance of harmonizing technical parameters when performing comparative analyses.

We further showcased the analytical utility of Immunemap by applying an unsupervised machine learning method to delineate recurrent motility patterns (Fig. 4). While previous studies either relied on manually defined categories or clustering with a predefined number of clusters (Dekkers et al, 2023), we employed a density-based approach combined with DTW to detect natural groupings in the data (Pizzagalli et al, 2019a). This method overcomes the limitations of convex clustering algorithms such as k-means and does not require a fixed number of clusters. Although the algorithm initially suggested eight primary clusters and several smaller subclusters, we selected the ten highest-scoring clusters based on the density × distance criterion (Pizzagalli et al, 2019a) to facilitate comparison with prior studies (Dekkers et al, 2023). This enabled the identification of distinct migratory behaviors—including directed movement, arrest, meandering, and alternating patterns—as well as a cluster indicative of tracking artifacts. While subtypes of meandering were previously described (Liew et al, 2017; Niemietz et al, 2024; Bonnardel et al, 2019), Immunemap enables future fine-grained analysis of subclusters along with their correlation with tissue type, immune cell identity, or perturbation to uncover context-specific dynamics.

A central advantage of Immunemap is its ability to integrate data across heterogeneous experimental conditions, allowing for large-scale analyses of immune cell behavior. For example, we used the platform to compare motility metrics such as speed, directionality, and arrest coefficient across various immune cell types (Fig. 3D–F), anatomical locations (Fig. 5F–H), and stimuli (Fig. 5C–E). These cross-dataset comparisons highlight the potential of Immunemap for hypothesis generation at scale. As in the case of other data-based initiatives, the contribution of new data by the community, particularly from steady-state conditions, will refine the resolution of these comparisons. This will enable researchers to systematically map how immune cell dynamics vary not only between inflammation and homeostasis, but also across tissue architectures, genetic models, or therapeutic interventions.

While Immunemap is designed to promote data reuse, it also contributes to standardization. The platform's upload workflow involves both human curation and automated quality scoring, fostering higher reporting standards in IVM. Each dataset is linked to rich metadata describing experimental design, imaging conditions, and mouse models, helping to harmonize downstream analyses. Additionally, Immunemap offers control datasets and encourages standardized protocols, which can serve as benchmarks for algorithm development and protocol refinement. These features also enable broader adoption of best practices across laboratories, reducing inter-study variability and increasing reproducibility.

Furthermore, Immunemap also supports the 3Rs—Replacement, Reduction, and Refinement—by enabling data reuse, reducing redundant animal experiments, and providing annotated examples for protocol optimization. As a public repository for validated, annotated IVM datasets, it creates opportunities for computational training, benchmarking, and educational use, reducing the need for repeated in vivo experiments.

Additionally, Immunemap's open-access design adheres to the FAIR principles—ensuring that data are findable, accessible, interoperable, and reusable. This fosters cross-disciplinary collaboration between immunologists, cell biologists, imaging experts, and data scientists. For instance, through publicly available APIs, users can retrieve data in HDF5, JSON, or CSV formats for custom analysis in R, Python, or Matlab. We demonstrated this by developing an interface with the CelltrackR R package (Wortel et al, 2021) to analyze motility metrics across thousands of tracks (Suppl. Material 1). Additionally, Immunemap supports applications beyond tracking, such as optical flow-based organ-level motility mapping (Appendix Fig. S5A–C) and training of centroid detection algorithms using deep learning (Appendix Fig. S5D,E).

In summary, Immunemap provides a foundational resource for the systematic, large-scale investigation of immune cell dynamics. By integrating curated imaging datasets, rich metadata, and computational tools in an open-access framework, it enables reproducible and data-driven insights into immune behavior across tissues and conditions. As the platform grows, it promises to catalyze new discoveries in immunology and quantitative bioimaging, while also serving as a bridge between biological research and computational science.

# Methods

**Reagents and tools table**

| Reagent/resource | Reference or source | Identifier or catalog number |
| --- | --- | --- |
| **Experimental models** | | |
| **Recombinant DNA** | | |
| **Antibodies** | | |
| **Oligonucleotides and other sequence-based reagents** | | |
| **Chemicals, enzymes and other reagents** | | |
| **Software** | | |
| Python 3.12.3 | https://www.python.org<br>Van Rossum, G. & Drake, F.L. Python 3 Reference Manual, 2009 | |
| Matlab 2024a | https://www.mathworks.com/products/matlab.html<br>MATLAB Release Notes, MathWorks, 2024 | |

| Reagent/resource | Reference or source | Identifier or catalog number |
|---|---|---|
| Dtaidistance for Dynamic Time Warping 2.3.13 | https://dtaidistance.readthedocs.io Wannes Meert, dtaidistance: Dynamic Time Warping Library, 2018 | |
| Matplotlib 3.10.0 | https://matplotlib.org Hunter, J.D., Matplotlib: A 2D Graphics Environment, Computing in Science & Engineering, 2007 | |
| pandas 2.2.3 | https://pandas.pydata.org McKinney, W., Data Structures for Statistical Computing in Python, Proceedings of the 9th Python in Science Conference, 2010 | |
| sklearn 1.4.2 | https://scikit-learn.org Pedregosa et al, Scikit-learn: Machine Learning in Python, Journal of Machine Learning Research, 2011 | |
| scipy 1.15.3 | https://scipy.org Virtanen et al, SciPy 1.0: Fundamental Algorithms for Scientific Computing in Python, Nature Methods, 2020 | |
| umap-learn 0.5.7 | https://umap-learn.readthedocs.io Leland McInnes, John Healy, James Melville, UMAP: Uniform Manifold Approximation and Projection, Journal of Open Source Software, 2018 | |
| Symfony 5.3 | https://symfony.com/ | |
| PHP 7.4 | https://php.net | |
| IMARIS 10.2.0 | https://imaris.oxinst.com Bitplane AG, Imaris Reference Manual (latest), 2024 | |
| ImageJ 1.54p | https://fiji.sc/ Schindelin, J., Arganda-Carreras, I., Frise, E., Kaynig, V., Longair, M., Pietzsch, T., ... Cardona, A. (2012). Fiji: an open-source platform for biological-image analysis. Nature Methods, 9(7), 676–682. https://doi.org/10.1038/nmeth.2019 | |
| Manual Tracking Plugin in FIJI 2.1.1 | https://imagej.net/ij/plugins/track/track.html | |
| Trackmate - January 2020 update | https://imagej.net/plugins/trackmate/ Tinevez, J.-Y., Perry, N., Schindelin, J., Hoopes, G. M., Reynolds, G. D., Laplantine, E., ... Eliceiri, K. W. (2017). TrackMate: An open and extensible platform for single-particle tracking. *Methods*, *115*, 80–90. https://doi.org/10.1016/j.ymeth.2016.09.016 | |
| Other | | |

## Methods and protocols

### Estimation of Z dimension and validation

To estimate the z-coordinates of cells from 2D tracks, we used the "Retrieve Z Coordinates" feature of the Manual Tracking plugin in FIJI. A Python script was then employed to extract the pixel intensity at each previously tracked 2D position and match it to its corresponding z-depth in the original 3D image stack.

Specifically, for each centroid in the track, the script identified the (x, y) coordinates in the 2D+time maximum intensity projection and retrieved the full 3D stack corresponding to that timepoint. The z-coordinate was estimated by locating the z-slice at which the intensity value at (x, y) matched the one observed in the 2D projection, assuming this corresponds to the plane of highest intensity at that location.

To validate this approach, we used a previously published dataset of manually annotated 3D tracks (Pizzagalli et al, 2018). We compared the estimated z values to the ground truth using the R-squared ($R^2$) metric, which quantifies the proportion of variance in the actual z values explained by the estimated ones. An $R^2$ value above 0.8 was considered indicative of accurate z estimation.

### Motility metrics and track similarity

For the analyses presented in Figs. 3 and 5, the following motility metrics were computed: speed, defined as the track length (mm) divided by track duration (min.), directionality, defined as the length of the vector connecting the first point and the last point of a track, divided by the track length (scalar from 0 to 1), square displacement: defined as the square of the difference between all the cell position at and its initial position, and arrest coefficient defined as the weighted sum of instantaneous speed—a speed threshold as defined previously (Pizzagalli et al, 2019b). These metrics were computed in 2 d + time using Matlab scripts except for results presented in Appendix Fig. S4, motility metrics that were computed with CelltrackR.

To identify clusters of tracks with similar motility patterns in Fig. 4, the measures described above were computed on 3D tracks, on a window of 50 s (corresponding to five time points). To avoid discretization of the metric values introduced by such a reduced temporal window, arrest coefficient was computed as the inverse of mean-squared displacement. Similarity of tracks was computed via Multivariate DTW, as previously demonstrated (Dekkers et al, 2023). This ensured the possibility to compare tracks with different durations. Different temporal sampling rate was handled by linear interpolation of centroid coordinates with a uniform sampling frequency (10 s). The features calculated from the sliding window approach created a four-dimensional time series for each track. These features were normalized and processed via DTW to get an $N \times N$ distance matrix ($N$ number of tracks). The matrix was reduced to a 2D space via UMAP (min_distance = 0.1, neighbors = 5). Metrics, DTW, and UMAP were computed through custom Python scripts, whereas clustering was done using Matlab.

For the analysis of the effect of temporal sampling rate on cell speed, arrest coefficient, and directionality presented in Fig. 5A,B, an exponential decay was fitted using the curve fitting toolbox of Matlab, with a curve of type "exp1".

### Unsupervised learning

To identify the main clusters of motility patterns displayed by immune cells, we employed a clustering method relying on density peaks and graph theory (Pizzagalli et al, 2019a). All the tracks uploaded to Immunemap by June 30, 2024, were included in the presented analysis. Clustering was performed by inspecting the density-distance plot and selecting ten main clusters (corresponding to ten density points with the highest density × distance score). Points were grouped into ten clusters using graph-based metrics for similarity (shortest path cost, pruning graph edges whose cost was

>0.1 to reduce computation time). To gain insight into the features associated with each cluster and compare them, we calculated the mean values of each feature within a cluster (min-max normalized before plotting) and presented them on a radar chart.

### Comparison of naïve CD8+ T cell motility after CRISPR-Cas9 engineering

We selected an experiment consisting of eight videos stored in Immunemap to compare the motility of CD8+ T cells in the popliteal lymph node after nucleofection-based CRISPR/Cas9 genetic engineering. Data were loaded into CelltrackR (v1.2.0 in R v4.2.3) for preprocessing and computing motility metrics (Appendix Fig. S4), including mean-squared displacement (MSD) and autocovariance. For the analyses, we selected three videos that contained freshly isolated control cells along with mock-nucleofected ("P4_mock") or nucleofected ("P4_DS137") cells. To further compare motility between the populations in these videos, we first extracted "tracklets" of equal length (20 steps with max 15 steps overlap). We then computed five motility metrics on each tracklet (speed, straightness, sphericity, outreach ratio, and mean *Turning Angle* in CelltrackR) and visualized the outcome in principal component analysis. To compare track speeds across all eight videos, we first corrected speeds based on the freshly isolated control population in all videos. While it is possible to pool the data across videos for analysis simply, this can be misleading when motility also varies from video to video, especially when comparing populations that do not come from the same video and/or the number of tracks is different. 'Video motility' can then become a confounder in the analysis. Therefore, we corrected the variability between videos by first subtracting from each measured track speed and the average speed of freshly isolated cells from the same video. Since the freshly isolated cells should be the same between videos, this allows correction for video-dependent motility effects. The remaining "residual speeds" can then be pooled and compared using, for example, an ANOVA (R function *aov*) with a post-hoc Tukey test (*TukeyHSD*). See also file tutorial_celltrackr_subtypesT.html in Suppl. Material 1 describes the entire process step by step.

### Platform development

Immunemap relies on the cloud-based architecture structured as illustrated in Appendix Fig. S3. A web application composed of a backend and a frontend provides data management and retrieval services. The backend is built on PHP 7.4 and Symfony 5.3, a framework for building scalable and flexible web applications. The frontend is developed with Angular, chosen for its data-binding features, dependency injection, and ecosystem that facilitates the creation of dynamic web applications. Moreover, angular modular design allows for a clear separation of concerns, making it easier to manage and update the user interface as the application evolves.

### Rendering

Image visualization relies on a dedicated engine designed for efficient processing and high-quality display of IVM images in common web browsers. To this end, RAW data stored in the HDF5 format (3D + time multichannel, with different bit depth) are converted, for visualization, to a sequence of JPEG images (2D + time). This happens, first by normalizing each channel to 8 bits (range adjusted to discard 1% of saturated pixels), followed by applying maximum intensity projection. One JPG image per time point and each acquisition channel are generated. The final rendered image is obtained by combining each single channel with a user-selectable lookup table that enables the adjustment of the color and the contrast.

### Quality metrics

Metrics to evaluate the quality of IVM videos were computed using Matlab (as shown in Fig. 4G). The Contrast Index quantifies the variance in pixel intensity between the original image and an automatically contrast-optimized reference image. The difference between the original and reference images is then normalized with respect to the reference. The Noise Index measures the difference in peak signal-to-noise ratio (PSNR) between the original and a reference image after applying a denoising median filter to the original image, again normalized to the reference. The Photo-bleaching Index assesses the decline in fluorescence intensity over time by calculating the slope of the linear interpolation derived from the mean intensities of individual frames. The Saturation Index evaluates the proportion of image pixels with uint8 values between 245 and 255. Lastly, the Signal Variation Index determines the standard deviation of image intensity across all frames of the videos.

### Usage notes

We recommend users retrieve data from Immunemap from multiple experiments to consider metadata and experimental details. It is particularly important to consider the effect of measurement bias that can be due to different acquisition settings, such as imaged volume or sampling rate, as shown in Appendix Fig. S4. The platform provides dedicated APIs to retrieve metadata and experimental settings to facilitate this process. Code examples for Matlab, Python, R, and documentation are provided in Suppl. Material 1.

## Data availability

The datasets produced in this study are available in the database Immunemap: https://www.immunemap.org.

The source data of this paper are collected in the following database record: biostudies:S-SCDT-10_1038-S44318-025-00629-4.

## Peer review information

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

## Acknowledgements

This work was financially supported by the Swiss National Science Foundation grants n.189699 and 204636, by swissuniversities.ch via the Swiss Open Research Data Grant CHORD-B, a FIR grant from USI and the support of the Helmut Horten Foundation.

## Author contributions

**Diego Ulisse Pizzagalli**: Supervision; Funding acquisition; Investigation; Writing—original draft. **Pau Carrillo-Barberà**: Investigation. **Himanshu Bansal**: Investigation; Writing—review and editing. **Elisa Palladino**: Investigation. **Kevin Ceni**: Investigation. **Benedikt Thelen**: Investigation. **Alain Pulfer**: Investigation. **Enrico Moscatello**: Investigation. **Raffaella Fiamma Cabini**: Investigation. **Johannes Textor**: Investigation. **Inge M N Wortel**: Investigation. **Rolf Krause**: Investigation. **Santiago Fernandez Gonzalez**: Supervision; Funding acquisition; Investigation; Writing—original draft; Writing—review and editing.

Source data underlying figure panels in this paper may have individual authorship assigned. Where available, figure panel/source data authorship is listed in the following database record: biostudies:S-SCDT-10_1038-S44318-025-00629-4.

## Disclosure and competing interests statement

The authors declare no competing interests.

## Immunemap Consortium

Michael Hickey[10], Ursula Norman[10], Andres Hidalgo[11], Georgiana Crainiciuc[11], Jose M Adrover[11,12], Miguel Palomino-Segura[11,13,14], Francesco Marangoni[15,16], Thorsten R Mempel[17], Chris Xu[18], Kibaek Choe[18], Ana-Maria Lennon-Dumenil[19], Dorian Obino[19], Philippe Bousso[20], Helene D Moreau[20], Cornelia Halin[21], Morgan Campbell Hunter[21], Jens V Stein[22], Petra Pfenninger[22], Jun Abe[22], Thomas T Murooka[23], Matteo Iannacone[24,25], Xenia Ficht[24,25], Federica Moalli[24,25], Alexandre P Bénéchet[24,25,26], Wolfgang Kastenmüller[27], Sarah Eickhoff[27,28], Milka Sarris[29], Antonios Georgantzoglou[29,30], Britta Engelhardt[31], Mykhailo Vladymyrov[31], Javier Pareja[31], Neda Haghayegh Jahromi[31], Michael D Cahalan[16], Shivashankar Othy[16], Yagmur Farsakoglu[32], Hans-Uwe Simon[33], Nina Germic[33], Mauro Di Pilato[34], Jordi Sintes[35], Tommaso Virgilio[1], Irene Latino[1], Daniel Molina Romero[1], Chiara Pizzichetti[1,36], Arianna Cappucetti[1], Kamil Chahine[1,36], Florentino Luciano Caetano dos Santos[1] & Joy Bordini[1]

[10]Centre for Inflammatory Diseases, Monash University, Melbourne, VIC, Australia. [11]Area of Cell & Developmental Biology, Centro Nacional de Investigaciones Cardiovasculares Carlos III, Madrid, Spain. [12]Cancer Macroenvironment Lab, The Francis Crick Institute, London, UK. [13]Department of Physiology, Faculty of Sciences, University of Extremadura, Badajoz, Spain. [14]Immunophysiology Research Group, Instituto Universitario de Investigación Biosanitaria de Extremadura (INUBE), Badajoz, Spain. [15]Institute for Immunology, University of California Irvine, Irvine, CA, USA. [16]Department of Physiology and Biophysics, University of California Irvine, Irvine, CA, USA. [17]Center for Immunology and Inflammatory Diseases, Division of Rheumatology, Allergy and Immunology, Massachusetts General Hospital and Harvard Medical School, Boston, MA, USA. [18]School of Applied and Engineering Physics, Cornell University, Ithaca, NY, USA. [19]Institut Curie, Centre de Recherche, PSL Research University, Paris, Île-de-France, France. [20]Institut Pasteur, Dynamics of Immune Responses Unit, Paris, Île-de-France, France. [21]ETH, Institute of Pharmaceutical Sciences, Zürich, Switzerland. [22]Department of Oncology, Microbiology and Immunology, University of Fribourg, Fribourg, Switzerland. [23]Department of Immunology, Rady Faculty of Health Sciences, University of Manitoba, Winnipeg, MB, Canada. [24]Division of Immunology, Transplantation, and Infectious Diseases, IRCCS San Raffaele Scientific Institute, Milan, Italy. [25]Vita-Salute San Raffaele University, Milan, Italy. [26]In Vivo Imaging Facility (IVIF), Department of Research and Training, Lausanne University Hospital and University of Lausanne, Lausanne, Switzerland. [27]Institute for Systems Immunology, University of Wurzburg, Wurzburg, Germany. [28]Institute of Experimental Oncology (IEO), Medical Faculty, University Hospital Bonn, University of Bonn, Bonn, Germany. [29]Department of Physiology, Development and Neuroscience, Downing Site, University of Cambridge, Cambridge, UK. [30]The Novo Nordisk Foundation Center for Stem Cell Medicine (reNEW), Department of Biomedical Sciences, University of Copenhagen, Copenhagen, Denmark. [31]Theodor Kocher Institute, University of Bern, Bern, Switzerland. [32]Department of Biomedicine, University of Basel, Basel, Switzerland. [33]Institute of Pharmacology, University of Bern, Bern, Switzerland. [34]Department of Immunology, The University of Texas MD Anderson Cancer Center, Houston, TX, USA. [35]Program for Inflammatory and Cardiovascular Disorders, Institut Hospital del Mar d'Investigacions Mediques (IMIM), Barcelona, Spain. [36]Graduate School for Cellular and Biomedical Sciences, University of Bern, Bern, Switzerland.

