## [Peer Review File · The EMBO Journal]

Systematic analysis of immune cell motility leveraging the open intravital microscopy database Immunemap

Diego Pizzagalli, Pau Carrillo-Barberà, Himanshu Bansal, Elisa Palladino, Kevin Ceni, Benedikt Thelen, Alain Pulfer, Enrico Moscatello, Raffaella Fiamma Cabini, Johannes Textor, Inge M. N. Wortel, Rolf Krause, and Santiago Fernandez Gonzalez

Corresponding author(s): Santiago Fernandez Gonzalez (santiago.gonzalez@irb.usi.ch) , Diego Pizzagalli (diego.pizzagalli@usi.ch)

Review Timeline:

Submission Date:	4th Feb 25
Editorial Decision:	13th Mar 25
Revision Received:	9th Jul 25
Editorial Decision:	18th Aug 25
Revision Received:	1st Sep 25
Accepted:	17th Oct 25

Editor: Ioannis Papaioannou

Transaction Report:

Dear Dr. Fernandez Gonzalez,

Thank you for submitting your manuscript EMBOJ-2025-120376 for consideration by The EMBO Journal, and for your patience during peer review. Your manuscript has now been seen by three experts in the field, and we have received the full set of their comments, which you can find below.

As you will see, all three referees find your collection of immune cell motility data in a publicly accessible and curated repository -significantly facilitating their accessibility and analysis- useful to the community and commendable. They all identify, however, several limitations in the analysis of these data presented in the current version of your manuscript, which together point to the conclusion that this part of the manuscript is not sufficiently developed for publication in The EMBO Journal.

We think that the suggestions of the referees regarding the analysis of the data and the discussion of the conclusions are largely reasonable and constructive, as well as realistically addressable in a single revision round, which is in line with our journal's policy.

Given the referees' positive comments and recommendations, and the importance of such an open-access repository for the community, I would like to invite you to submit a thoroughly revised version of your manuscript along with a detailed point-by-point response addressing all referees' comments. I should add that it is The EMBO Journal policy to allow only a single round of major revision, and acceptance of your manuscript will therefore depend on the completeness of your responses in this revised version. Please let me know if you have any questions or comments that you would like to discuss with me.

We generally allow three months as standard revision time (June 12, 2025). As a matter of policy, competing manuscripts published during this period will not negatively impact our assessment of the conceptual advance presented by your study. However, we request that you contact us as soon as possible upon publication of any related work, to discuss how to proceed. Should you foresee a problem in meeting this three-month deadline, please let us know in advance and we may be able to grant an extension.

Thank you for the opportunity to consider your work for publication in The EMBO Journal. I look forward to your revision.

Best regards,

Ioannis

Instructions for preparing your revised manuscript

1. When you are ready to submit the revision, please upload:

- A Word file of the manuscript text (including legends of main Figures, EV Figures and Tables). Please make sure that changes are highlighted (or "tracked") to be clearly visible.

- Individual production-quality figure files (one file per figure). When assembling your figures, please refer to our figure preparation guidelines in order to ensure proper formatting and readability in print as well as on screen:

If the data shown in a figure are obtained from n {less than or equal to} 2, please use scatter plots showing the individual data points.

- i. the name of the statistical test used to generate error bars and P values
- ii. the number (n) of independent experiments (please specify technical or biological replicates) underlying each data point (discussion of statistical methodology can be reported in the Materials and Methods section, but figure legends should contain a basic description of n , P , and the test applied)
- iii. the nature of the bars and error bars (s.d., s.e.m.).

- A point-by-point response to the referees' comments, with a detailed description of the changes made (as a word file). All referees' concerns must be fully addressed and their suggestions taken on board. When preparing your letter of response to the referees' comments, please bear in mind that this will form part of the Review Process File and will therefore be available online to the community. Please note that you have the possibility to opt out of the transparent process at any stage prior to publication by letting the editorial office know (contact@embojournal.org); if you do opt out, the Review Process File link will point to the following statement: "No Review Process File is available with this article, as the authors have chosen not to make the review process public in this case.". For more details on our Transparent Editorial Process, please visit our website: <https://www.embopress.org/page/journal/14602075/authorguide#transparentprocess>

- Expanded View (EV) files (replacing Supplementary Information) that are collapsible/expandable online. A maximum of 5 EV Figures can be typeset. EV Figures should be cited as "Figure EV1, Figure EV2" etc. in the text, and their respective legends should be included in the manuscript file after the legends of regular figures. See detailed instructions regarding Expanded View files here: <https://www.embopress.org/page/journal/14602075/authorguide#expandedview>

- For the figures that you do NOT wish to display as Expanded View figures, they should be bundled together with their legends in a single PDF file called "Appendix", which should start with a short Table of Contents (including page numbers). Appendix figures should be referred to in the main text as: "Appendix Figure S1, Appendix Figure S2" etc. Please see detailed instructions here: <https://www.embopress.org/page/journal/14602075/authorguide#expandedview>

- A complete author checklist, which you can download from our author guidelines (<https://www.embopress.org/page/journal/14602075/authorguide>). Please note that the checklist will also be part of the Review Process File.

2. Please note that no statistics should be calculated and shown in Figures if $n=2$. Please also note that each p value should be reported as an exact value.

3. Before submitting your revision, primary datasets (and computer code, where appropriate) produced in this study need to be deposited in appropriate public databases (see <https://www.embopress.org/page/journal/14602075/authorguide#dataavailability>). The accession numbers, database, and the specific URLs (links) should be listed in a formal "Data availability" section (placed after Methods), following the example below:

"The RNA-seq datasets produced in this study are available in the following database:
Gene Expression Omnibus GSE46843 (<https://www.ncbi.nlm.nih.gov/geo/query/acc.cgi?acc=GSE46843>)"

*** All links should resolve to a page where the data can be accessed. ***

*** Please remember to provide in the Data availability section of your revised manuscript reviewer passwords if the datasets are not yet public. ***

*** The Data Availability Section is restricted to new primary data that are part of this study. In case you have no data that require deposition in a public database, please state so instead of referring to the database: "Our study includes no data deposited in public repositories." under the heading "Data availability". ***

4. The materials and methods need to be described in the manuscript using our structured methods format, which is now required for all research articles. According to this format, the Methods section includes a single "Reagents and Tools Table" - listing key reagents, experimental models, software and relevant equipment including their sources and relevant identifiers - followed by a "Methods and Protocols" section describing the methods. Please download and fill our Reagents and Tools Table template (.docx), which you can find in our author guide: <https://www.embopress.org/page/journal/14602075/authorguide#structuredmethods>. When submitting your revised manuscript, please do not include the Reagents and Tools Table in the Methods section of the manuscript but instead upload it as a separate file choosing the file type "Reagent Table".

5. Please check that the title and the abstract of the manuscript are brief, yet explicit, even to non-specialists. The length of the title should not exceed 100 characters, and the abstract should be a single paragraph not exceeding 175 words.

6. Please also note our reference format: <https://www.embopress.org/page/journal/14602075/authorguide#referencesformat>.

8. Please remember: digital image enhancement is acceptable practice, as long as it accurately represents the original data and conforms to community standards. If a figure has been subjected to significant electronic manipulation, this must be noted in the figure legend or in the "Materials and Methods" section. The editors reserve the right to request original versions of figures and the original images that were used to assemble the figure.

9. Our journal encourages inclusion of data citations in the reference list to directly cite datasets that were obtained from public databases. Data citations in the article text are distinct from normal bibliographical citations and should directly link to the database records from which the data can be accessed. In the main text, data citations are formatted as follows: "Data ref: Smith et al, 2001" or "Data ref: NCBI Sequence Read Archive PRJNA342805, 2017". In the Reference list, data citations must be labeled with "[DATASET]". A data reference must provide the database name, accession number/identifiers, and a resolvable link to the landing page from which the data can be accessed at the end of the reference. Further instructions are available at: <https://www.embopress.org/page/journal/14602075/authorguide#referencesformat>.

10. We request authors to consider both actual and perceived competing interests. Please review our policy (<https://www.embopress.org/page/journal/14602075/authorguide#conflictsofinterest>) and update your competing interests statement if necessary. Please name this section 'Disclosure and competing interests statement' and place it after the Acknowledgements section.

11. Please note that all corresponding authors are required to provide an ORCID ID upon submission of a revised manuscript (<https://orcid.org/>). Please find instructions on how to link your ORCID ID to your account in our manuscript tracking system in our Author guidelines (<https://www.embopress.org/page/journal/14602075/authorguide#authorshipguidelines>).

12. We use CRediT to specify the contributions of each author in the journal submission system. CRediT replaces the author contribution section, which should be removed from the manuscript. Please use the free text box to provide more detailed descriptions. See also guide to authors: <https://www.embopress.org/page/journal/14602075/authorguide#authorshipguidelines>.

14. We would also welcome the submission of cover suggestions or motifs to be used by our Graphics Illustrator in designing a cover.

15. Please use the link below to submit your revision:
<https://emboj.msubmit.net/cgi-bin/main.plex>

Referee #1:

In this manuscript, Pizzagalli et al. present Immunemap as a collection of immune cell motility datasets acquired using intravital microscopy contributed by a consortium of international research groups.

This initiative provides a publicly accessible repository that hosts not only the original video files, but in many cases manually annotated cell tracks that further enable further downstream analyses by researchers accessing the datasets. The implementation of the in-browser visualization tool is both adequate and user-friendly, but the documentation that should detail how to best leverage such a vast amount of data remains incomplete (see comments below).

The authors also conducted analyses based on nearly 17000 cell tracks pooled from the datasets to assess the motility patterns of cells across different tissues, experimental conditions and cell types. However, certain aspects of the analyses are flawed and could undermine the interpretability of the analyzed data. Additionally, several figures mentioned in the manuscript were also not provided/presented.

Major

1. All cell tracks annotated and analyzed in this manuscript come from using 2d + time representations instead of the original 3d volume + time movie data (such as the illustration shown in Fig. 1B). One example of this transformation is shown in Fig. 3C from this experiment: <https://app.immunemap.org/experiment-public-view?id=48> where the cells in the original videos, as is true of most intravital microscopy videos, were moving in 3d space.

This important omission was only briefly stated in the Methods section under Motility metrics (line 423) but was not mentioned at

all in the main text.

Given that cells move in 3d space, it is imperative that the derived metrics such as speed, arrest coefficient and directionality from the cell tracks encompass all three spatial dimensions, as the missing dimension could substantially alter the profile of the cell tracks. As just one example of this problem, a rapidly moving cell migrating along the vertical z-axis would be (incorrectly) registered as stationary and having low motility.

This reviewer understands that manual tracking of the cells in hundreds of movies can be very time consuming and tedious. However, this critical limitation needs to be clearly stated in the main text such that the audience understands that the derived parameters were conducted on cell tracks "flattened" on to a 2d "plane" and thus could be missing potentially useful information from the omitted dimension.

2. Fig. 4C: Unsupervised clustering based on density peaks and graph theory.

The authors did not provide sufficient details in the Methods section to reproduce the analysis. In particular, based on the referenced paper from the same first author (Pizzagalli et al., Sci. Adv., 2019: 10.1126/sciadv.aax3770), a density-distance plot (as described by Rodriguez and Laio, Science, 2014: 10.1126/science.1242072) was supposedly used to identify the density peaks.

As visual inspection of Fig. 4C gives the impression of a relatively contiguous scattering of the cells based on their arrest coefficient and directionality, rather than aggregates of dense clusters (it is not apparent that the four color-coded clusters center on any density peaks), it would be useful for the authors to show what the distance-density plots look like and which points in Fig. 4C correspond to those density peaks.

Furthermore, in the Methods section (lines 439-442), the authors mentioned that six density points were found and the points were grouped into six clusters, yet only four clusters of cells were shown in the analyses in Fig. 4C-H. Please clarify how and why the two additional clusters were trimmed/missing from the final analysis.

3. Considering that the scatter plot in Fig. 4C shows a rather contiguous mapping of cells across both parameters rather than distinct aggregates, this reviewer questions whether it would be accurate to describe them in four such distinct behaviors as "arrested", "focused patrolling", "extended patrolling" and "directed". It would be more accurate to describe them as high arrest coefficient, high directionality; low arrest coefficient, high directionality etc. and further explain that these distinct behaviors can be observed within these groups.

Additionally, as the values of arrest coefficient and directionality were derived from the ~500s tracklets averaged across all time points, it is unclear if the four distinct motility patterns could be "distilled" from these averaged values without extracting information on a frame-by-frame or subdividing the tracklets into smaller segments.

The authors should justify their claim that the classification of cell movement into the four distinct behaviors based on arrest coefficient and directionality alone is an adequate and accurate description for the majority of the cells analyzed in this manuscript.

4. Supplementary Materials 1 as mentioned in lines 240, 268 are supposed to contain the code examples and documentation but were not provided. As such, this reviewer has not been able to take advantage of the Immunemap repository to evaluate the usability/accessibility of the database.

It is strongly recommended that the authors provide robust documentation by including basic tutorials and demo examples such that the vast amount of data/information hosted on Immunemap could be utilized by other researchers. The current documentation (<https://immunemap.org/index.php/doc>) only provided very basic information on how to obtain the data from Immunemap via HTTP protocol, but does not further provide the steps (e.g. Python code) to convert the requested data (e.g., cell tracks) in JSON format into arrays (e.g. Numpy arrays in Python) that can be used for downstream analysis. This requires a certain level of programming experience from the users and could limit accessibility of the Immunemap repository.

5. Supplementary Figures 5 and 6 (mentioned in lines 240-244) are also missing.

6. In the Methods section, under the section Motility metrics (lines 417, 424), Fig. 6 was mentioned but this figure was not presented in the manuscript nor was it described anywhere in the main text.

Minor

1. Legend for Fig. 1C (3) spleen - green cells?

2. In Fig. 4G, under the "T cell activation" condition, the figure legend described the T cells as being stimulated by polyIC. Was this performed on antigen-specific T cells (if so, were they CD4+ or CD8+ T cells? Naïve or activated?) stimulated with their cognate antigens (if so, in what form, e.g. peptide pulsing?) or were they merely naïve non-specific T cells that underwent change in migratory behavior when polyIC was introduced into the tissues (which tissue?) without the presence of cognate

antigen? The context is important for interpreting the data presented in this subfigure.

Referee #2:

The study by Pizzagalli, Carrillo-Barbera et al. leverages over 400 intravital videos to create Immunemap, a novel atlas of immune cell motility. The authors brought together an impressive initiative of experts in the field of IVM to provide an open data platform for data-driven investigations. The study thus presents a fantastic collaborative effort and enhances the findability, accessibility and interoperability of IVM data. Importantly, the authors use opensource software for image analysis thus allowing reproducibility and open access. The real work lies within the data collection, generation of data infrastructure and curation of data. However, in its current form, the authors do not make use of the wealth of data that they have collected and the analyses presented seem rather superficial. In addition, some findings are overstated and need to be either solidified or worded differently. Here are my comments:

Major:

- Migration patterns: The authors claim that they identified two novel patterns of immune cell migration, namely focused patrolling and movement over larger areas. However this is clearly an overstatement as patrolling of focused areas and larger patrolling have been described numerous times (PMID: 29045904 Liew et al Immunity 2017, PMID: 38231043 Niemi et al. Hepatology 2024, PMID: 31561945 Bonnardel et al. Immunity 2019). It is unclear how these novel patterns were identified and deemed as novel. Please explain clearly or remove the claim of novelty because numerous migration patterns have been identified by the IVM community.
- Organ specificity: the authors bulk immune cell dynamics together without accounting for potential differences of the imaged organs. It would be important to understand if a neutrophil or CD4 T cell behaves differently in different organs. Could the authors perform analyses combining speed, directionality or arrest between organs and conditions? I do not think it makes sense to combine migration patterns of immune cells between different conditions. It would be better to stratify by organ and condition.
- Detail: the authors often neglect the description of details necessary to understand major findings. E.g. the identification of DCs, which lack a lineage marker and need a combination of markers, it is unclear how these are identified (e.g. Fig 3).
- Imaging modalities: a big issue in IVM is the use of different imaging platforms such as upright / inverted microscopes, 2-photon vs confocal, camera-based vs. laser scanning etc. Could the authors provide more details how this affected their analysis and is there a way to use computational analysis to harmonize data obtained with different imaging tools?

Minor:

- Fig. 2J: the intention is to compare a video with loss of fluorescence (left) to a stable one (right). However, the right video has either loss of focus or bleaching as visible by the disappearance of green autofluorescence in the last sequence.

Referee #3:

The manuscript by Hickey et al curated a large database of single cell motility data derived from intravital imaging analysis in murine models. The team manually tracked and annotated over 58k single cell trajectories from 400 videos taken under different disease/treatment conditions. Based on these extensive single cell motility data, quantitative and unsupervised clustering analysis were performed to identify distinctive phenotypes of immune cell motility. The effort and care in curating this large, in vivo immune cell motility database are surely commendable, as the research community indeed needs such a database. However, significantly more quantitative analysis of the annotated single cell tracks should be performed, especially considering that most of the single cell tracks are short, with only about 20 data points. Also, it will be very helpful and facilitate further design of intravital imaging experiments if the authors can analyze the data in a tissue and disease/treatment specific manner, instead of mostly lumping all the data together. The variations shown in Figure 3D-F are very large, indicating that it's not biologically meaningful to lump all the data together. Moreover, the textual discussions of the data analysis results should be substantially expanded so that the readers can know more about the biological insights derived from the data as well as the strengths and limitations of this intravital imaging dataset.

Major points:

1. Figure 3D-F: The standard deviations shown in these figures are very large for all immune cell types. To me, this indicates that the immune cells probably show different motility profiles, depending on the tissues and disease/treatment conditions. In Figure 3G, CD8 T cells clearly showed 2 distinctive clusters, suggesting two populations with very different motility characteristics. The authors should perform more detailed analysis of the immune cell motility specific to tissues and disease/treatment conditions. For example, it's interesting to even just know the immune cell motility in different tissues under the control condition (no disease, no treatment). Is T cell motility pretty much the same in lung and liver? What are the two populations of CD8+ T cells corresponding to? Significantly more data analysis are needed in order to truly understand the biological meaning of the data.
2. A related question is how reproducible the intravital imaging measurements are. The authors curated data from 20 labs. It will

be informative to know that data from these different labs are consistent and can indeed be grouped together to generate the database. A simple quality check is to compare single cell motility data of the same immune cell type under the same treatment condition, but acquired by different labs. For the 400 videos, are there such videos from different labs, i.e., with the same tissue model and treatment condition? If data from the different labs are consistent, it is reassuring for researchers that may want to use the database for further analysis. If the data from different labs are not consistent, it'll be helpful if the authors can identify possible origins of the experimental variability so that researchers can try to avoid such experimental variability when they design future intravital imaging experiments.

3. Figure 4C: Unsupervised clustering is used to identify the unique clusters. In the Materials and Methods section, it's stated that 6 clusters were identified. But only 4 were shown in Figure 4C. What are the 2 clusters that are not shown? Can the authors perform principal component analysis (PCA) on the dataset and see whether the data would cluster differently when using, e.g., the 1st and 2nd PC to visualize the data?

4. Have the authors calculated the Mean Squared Displacement (MSD) for the individual single cell tracks shown in Figure 4? A directionality index of 0.4-0.5 is not that large that would indicate non-random motion, while directionality index of 1 surely indicates directed motion. Grouping single cells with directionality of 0.4 with those close to 1 is not mechanistically meaningful to me. The authors should calculate and use more statistical motility parameters to characterize the single cell tracks (e.g., MSD) and perform the clustering analysis, especially considering the single cell tracks are mostly short.

5. Figure 4E: Based on the clustering analysis, CD4 T cells seem to be the most motile with >50% showing low arrest coefficient and high directionality (i.e., persistent motion). This is different from the conclusion of CD8 T cells being most motile drawn from Figure 3D-F. Is this because the dataset used to derive Figure 4E is smaller than that used for Figure 3? Again, it's important and informative to know the biological origins or correlation of the different motility phenotype distribution for the different immune cell types shown in Figure 4E. For instance, for the CD4 T cell tracks showing directed motion (denoted in green), are they mostly from the lymph nodes, while the arrest tracks are mostly from lung/liver (tissue-resident T cells)? The fact that the dataset is heavily biased towards lymph node data (Fig. 1D) makes such distribution analysis results not that meaningful because it's not really reflecting the motility phenotype distribution in vivo. It's more likely the result of data biased toward certain tissue and treatment condition. The authors should perform more careful analysis and take into account the highly heterogeneous numbers of data from the different tissues and treatment conditions (Fig. 1D) when presenting the data and drawing conclusions in Figure 4.

6. Figure 4F-H: This is very interesting and shows the biological insight that the database can provide. I would suggested the authors perform such analysis for all different immune cell types by comparing the control with the disease/treatment condition, and identify how the motility phenotypes may correspond to physiological or pathological processes. To me, this is one of the most impactful achievements of the database. It will also enable researchers to use the database for further quantitative analysis or as references. I understand this could involve a lot of figures. The authors can only show a few representative ones in the main text. The rest can be included as supplementary figures. The authors should also provide more discussions of the analysis results in the main text regarding the biological insight derived from the dataset.

Dear Dr. Papaioannou,

Let me initially apologize for the extra time required to complete all the requests made by the revision process. We would also like to thank all the reviewers for their insightful comments. We have modified the manuscript according to their suggestions. Amongst other actions, we have included the addition of 3D track information, the re-analysis of the clustering using a novel algorithm, and the evaluation of different parameters that might affect the reproducibility of the data, including the comparison between organs and conditions. We do consider that all these additions have greatly improved the manuscript and we think that this work will be an excellent resource for the research working in the immunology field as well as in the *in vivo* imaging community.

We have included in this letter a point-by-point answer to all the comments from the reviewers, indicating the added parts in the revised manuscript.

Best regards,

Santiago Gonzalez.

Referee

#1:

In this manuscript, Pizzagalli et al. present Immunemap as a collection of immune cell motility datasets acquired using intravital microscopy contributed by a consortium of international research groups.

This initiative provides a publicly accessible repository that hosts not only the original video files, but in many cases manually annotated cell tracks that further enable further downstream analyses by researchers accessing the datasets. The implementation of the in-browser visualization tool is both adequate and user-friendly, but the documentation that should detail how to best leverage such a vast amount of data remains incomplete (see comments below).

The authors also conducted analyses based on nearly 17000 cell tracks pooled from the datasets to assess the motility patterns of cells across different tissues, experimental conditions and cell types. However, certain aspects of the analyses are flawed and could undermine the interpretability of the analyzed data. Additionally, several figures mentioned in the manuscript were also not provided/presented.

Major

1. All cell tracks annotated and analyzed in this manuscript come from using 2d + time representations instead of the original 3d volume + time movie data (such as the illustration shown in Fig. 1B). One example of this transformation is shown in Fig. 3C from this experiment: <https://app.immunemap.org/experiment-public-view?id=48> where the cells in the original videos, as is true of most intravital microscopy videos, were moving in 3d space.

This important omission was only briefly stated in the Methods section under Motility metrics (line 423) but was not mentioned at all in the main text.

Given that cells move in 3d space, it is imperative that the derived metrics such as speed, arrest coefficient and directionality from the cell tracks encompass all three spatial dimensions, as the missing dimension could substantially alter the profile of the cell tracks. As just one example of this problem, a rapidly moving cell migrating along the vertical z-axis would be (incorrectly) registered as stationary and having low motility.

This reviewer understands that manual tracking of the cells in hundreds of movies can be very time consuming and tedious. However, this critical limitation needs to be clearly stated in the main text such that the audience understands that the derived parameters were conducted on cell tracks "flattened" onto a 2d "plane" and thus could be missing potentially useful information from the omitted dimension.

We thank the reviewer for the insightful comment. As noted, we chose to perform 2D cell tracking primarily because 3D tracking demands considerable time and effort (especially given the large number of videos and the need for manual annotation by multiple operators). Many dynamic cellular behaviours (including migration patterns, directional persistence, and interactions with surrounding structures) occur predominantly within tissue planes that can be effectively captured in 2D projections. Moreover, 2D tracking has been widely accepted as the standard approach in the majority of published studies investigating immune cell motility. However, as the reviewer pointed out, this methodology introduces a level of error that is often tolerated in favor of experimental scalability.

In response to the reviewer's suggestion, we analyzed how converting 3D trajectories to 2D affected key cell motility parameters, including speed, displacement, directionality or arrest coefficient. To do this, we used a subset of videos from a previous work (Pizzagalli et al. 2018), in which cells had been previously tracked in 3D. Our analysis revealed 3D-to-2D transformation can introduce errors across most of the evaluated parameters, for instance, speed and displacement were underestimated when measured in 2D, while arrest coefficient was overestimated. We believe these findings are important for the imaging community. Therefore, we have included them as part of the Revised Suppl. Fig. 2. In addition, to illustrate the impact more clearly, we identified and highlighted an example of a cell that predominantly moves along the z-axis. We compared this cell's 2D and 3D tracks to visually demonstrate the discrepancies introduced by dimensional reduction in that specific case. This example is also included in the Revised Suppl. Fig. 2A.

Therefore, we decided to integrate the z dimension back into the collection of tracks from Immunemap. To achieve this, we apply the "Retrieve z coordinate" feature from the manual tracking plugin from Fiji that uses the maximum intensity projection to estimate the z plane (<https://imagej.net/ij/plugins/track/track.html>). In addition, we validated the algorithm using the aforementioned collection of 3D-tracked videos (Pizagalli et al., 2018), confirming its high accuracy in estimating the z-coordinate. Specifically, the coefficient of determination (R^2) exceeded 0.8 for the majority of the tested tracks (Revised Suppl. Fig. 2K).

Supplementary Figure 2. Impact of dimensionality reduction on key motility metrics. *A*) Graphic representation of a representative 3D immune cell trajectory (blue) and its 2D-projected counterpart (red), generated by collapsing the z-axis. **B-I**) Quantitative comparison of motility features computed from the original 3D track versus its 2D projection (track: LTDB013_a_GT_2, from the LTDB dataset²⁹). Metrics shown include square displacement (**B, C**), mean speed (**D, E**), arrest coefficient (**F, G**), and directionality (**H, I**). For each feature, values calculated in 2D (excluding z-axis) are compared with those from the full 3D track (including z-axis). The panels labeled "3D-2D" represent the difference between 3D and 2D values, calculated as: 3D metric – 2D metric, such that negative values indicate overestimation of the metric in 2D relative to 3D. **K**) Distribution of the R² coefficient among 3D videos, showing an average of 0.84.

2. Fig. 4C: Unsupervised clustering based on density peaks and graph theory. The authors did not provide sufficient details in the Methods section to reproduce the

analysis. In particular, based on the referenced paper from the same first author (Pizzagalli et al., *Sci. Adv.*, 2019: 10.1126/sciadv.aax3770), a density-distance plot (as described by Rodriguez and Laio, *Science*, 2014: 10.1126/science.1242072) was supposedly used to identify the density peaks.

As visual inspection of Fig. 4C gives the impression of a relatively contiguous scattering of the cells based on their arrest coefficient and directionality, rather than aggregates of dense clusters (it is not apparent that the four color-coded clusters center on any density peaks), it would be useful for the authors to show what the distance-density plots look like and which points in Fig. 4C correspond to those density peaks.

Furthermore, in the Methods section (lines 439-442), the authors mentioned that six density points were found and the points were grouped into six clusters, yet only four clusters of cells were shown in the analyses in Fig. 4C-H. Please clarify how and why the two additional clusters were trimmed/missing from the final analysis.

We agree with the reviewer that the motility patterns can look continuous in the space determined by the selected parameters and adjusted their definition. Therefore we performed a new analysis where the result cluster boundaries are more evident. To generate the Revised Fig. 4 using the new dataset that includes the 3D track information, we used a clustering analysis pipeline recently described by Dekkers and colleagues (2023). Briefly, this method consists of clustering the tracks based on their features using the dynamic time warping algorithm (DTW) which allows to compare the tracks as time series datasets. Following the reviewers suggestions, we have described this analysis in a detailed protocol in the Results section (Lines 227-266) and the Material and Methods section (Lines 483-491).

3. Considering that the scatter plot in Fig. 4C shows a rather contiguous mapping of cells across both parameters rather than distinct aggregates, this reviewer questions whether it would be accurate to describe them in four such distinct behaviors as "arrested", "focused patrolling", "extended patrolling" and "directed". It would be more accurate to describe them as high arrest coefficient, high directionality; low arrest coefficient, high directionality etc. and further explain that these distinct behaviors can be observed within these groups.

We agree with the reviewer's comment. The results obtained from the new clustering analysis allowed us to identify ten distinct clusters by combining parameters associated with cell motility (Revised Fig. 4A-D). Following the suggestion of the reviewers, we have identified each cluster according to their arrest coefficient, directionality, speed and displacement. as indicated in the Revised Figure 4B and represent them as a radar plot. In addition, we have described the clusters in the results section, (Lines 229-279), highlighting a continuous partitioning especially of the previously named "patrolling" clusters. In addition we have included a discussion paragraph regarding this topic (Lines 385-398).

Figure 4. Characterization of the main motility patterns displayed by immune cells via dynamic time warping and unsupervised learning. **A)** Scatter plot showing the similarity in cell trajectories calculated by dynamic time warping (DTW) in a UMAP-reduced dimensionality space. Colors represent the ten clusters attributed via density peak clustering. **B)** Radar plot showing motility properties associated with each cluster. Representative cluster features included: Cluster 1, consistent with directed migration; Cluster 10, largely arrested cells; Cluster 7, alternating phases of movement and arrest; Cluster 5, long meandering trajectories; and Cluster 6, likely tracking artefacts due to unusually high speed and displacement. The remaining clusters reflected variations of patrolling behavior with differing levels of directionality. Values indicate the average metric over all the tracks included in each cluster normalized using the min max normalization for radar chart interpretability. **C)** 3D cell tracks representative for each cluster. **D)** Distribution of the ten motility clusters among the different immune cell types tracked in Immunemap.

Additionally, as the values of arrest coefficient and directionality were derived from the ~500s tracklets averaged across all time points, it is unclear if the four distinct motility patterns could be "distilled" from these averaged values without extracting information on a frame-by-frame or subdividing the tracklets into smaller segments. The authors should justify their claim that the classification of cell movement into the four distinct behaviors based on arrest coefficient and directionality alone is an adequate and accurate description for the majority of the cells analyzed in this manuscript.

We previously used a window of 500s to compare fragments of tracks with the same duration. In this revised version we modified the analysis using Dynamic Time Warping with a window size reduced to 50s. Despite the smaller support, this sliding window is then applied to the entire track and averages values are computed. We explained this better in Materials and Methods (Lines 473-484). Hence, The new clustering approach compares the whole tracks instead of tracklets using the DTW algorithm. Since features such as Arrest Coefficient and Directionality cannot be computed on a frame-by-frame basis, we calculated them over a small sliding window of 50 seconds, which provides a finer temporal resolution suitable for capturing subtle cell movements.

4. Supplementary Materials 1 as mentioned in lines 240, 268 are supposed to contain the code examples and documentation but were not provided. As such, this reviewer has not been able to take advantage of the Immunemap repository to evaluate the usability/accessibility of the database.

It is strongly recommended that the authors provide robust documentation by including basic tutorials and demo examples such that the vast amount of data/information hosted on Immunemap could be utilized by other researchers. The current documentation (<https://immunemap.org/index.php/doc>) only provided very basic information on how to obtain the data from Immunemap via HTTP protocol, but does not further provide the steps (e.g. Python code) to convert the requested data (e.g., cell tracks) in JSON format into arrays (e.g. Numpy arrays in Python) that can be used for downstream analysis. This requires a certain level of programming experience from the users and could limit accessibility of the Immunemap repository.

In this revised version we provide documentation and supplementary files. This includes description of the HTTP, APIs and code examples in Matlab, Python, R. See Supplementary Material 1.

5. Supplementary Figures 5 and 6 (mentioned in lines 240-244) are also missing.

We apologize for the mistake in the previously uploaded version of the manuscript. The missing figures have now been uploaded and updated (Revised Supplementary Fig. 5, Revised Fig. 5).

6. In the Methods section, under the section Motility metrics (lines 417, 424), Fig. 6 was mentioned but this figure was not presented in the manuscript nor was it described anywhere in the main text.

We apologize for the typo. In this revised version we corrected the mistakes in the Figure numbering. We described motility metrics in material and methods (Lines 464 - 489)

Minor

1. Legend for Fig. 1C (3) spleen - green cells?

The legend for Figure 1C (3) spleen was incorrect. We have now corrected this issue, and an Revised version of Figure 1 has been included in the revised manuscript.

2. In Fig. 4G, under the "T cell activation" condition, the figure legend described the T cells as being stimulated by polyIC. Was this performed on antigen-specific T cells (if so, were they CD4+ or CD8+ T cells? Naïve or activated?) stimulated with their cognate antigens (if so, in what form, e.g. peptide pulsing?) or were they merely naïve non-specific T cells that underwent change in migratory behavior when polyIC was introduced into the tissues (which tissue?) without the presence of cognate antigen? The context is important for interpreting the data presented in this subfigure.

We apologize for the missing details. This analysis refers to OT-I CD8+ T cells (Specific TCR for OVA) captured via 2 photon microscopy in the spleen under polyIC induced inflammation. We added this fragment at line 774 in the legend of the Revised Fig.5 (panel D) and in the Results (Line 293).

Referee

#2:

The study by Pizzagalli, Carrillo-Barbera et al. leverages over 400 intravital videos to create Immunemap, a novel atlas of immune cell motility. The authors brought together an impressive initiative of experts in the field of IVM to provide an open data platform for data-driven investigations. The study thus presents a fantastic collaborative effort and enhances the findability, accessibility and interoperability of IVM data. Importantly, the authors use opensource software for image analysis thus allowing reproducibility and open access. The real work lies within the data collection, generation of data infrastructure and curation of data. However, in its current form, the authors do not make use of the wealth of data that they have collected and the analyses presented seem rather superficial. In addition, some findings are overstated and need to be either solidified or worded differently. Here are my comments:

Major:

- Migration patterns: The authors claim that they identified two novel patterns of immune cell migration, namely focused patrolling and movement over larger areas. However this is clearly an overstatement as patrolling of focused areas and larger patrolling have been described numerous times (PMID: 29045904 Liew et al Immunity 2017, PMID: 38231043 Niemietz et al. Hepatology 2024, PMID: 31561945 Bonnardel et al. Immunity 2019). It is unclear how these novel patterns were identified and deemed as novel. Please explain clearly or remove the claim of novelty because numerous migration patterns have been identified by the IVM community.

We thank the reviewer for pointing out the overstatement regarding novelty. We agree that the distinction between focused and extended patrolling has already been described in the literature (e.g., Liew et al. 2017, Bonnardel et al. 2019, Niemietz et al. 2024). However, our intention was primarily technical—to emphasize that these patterns were initially identified through unsupervised learning. In the revised manuscript, we have removed any claims of novelty regarding the migration patterns and instead present the analysis as demonstration of the types of insights that can be generated using Immunemap. Moreover, we reanalyzed the data using dynamic time warping (DTW) followed by clustering as previously implemented by Dekker et al. (2023). This approach allowed us to identify and describe a range of migration behaviors that emerge in an unsupervised manner from aggregated trajectories (Revised Fig. 4). These categories are more continuous and do not discretize cell behavior into distinct categories. Our goal with this analysis is not to introduce new categories of migration but to demonstrate how Immunemap enables systematic, data-driven identification and comparison of cell movement patterns across heterogeneous datasets. We clarified this point in the revised manuscript and rephrased the relevant sections accordingly.

- Organ specificity: the authors bulk immune cell dynamics together without accounting for potential differences of the imaged organs. It would be important to understand if a neutrophil or CD4 T cell behaves differently in different organs. Could the authors perform analyses combining speed, directionality or arrest between organs and conditions? I do not think it makes sense to combine migration patterns of immune cells between different conditions. It would be better to stratify by organ and condition.

We thank the reviewer for the comment. The comparison of motility patterns across different organs is indeed a valuable and biologically relevant aspect of Immunemap. As suggested, analyzing cell motility under steady-state conditions would likely offer the clearest insight into organ-specific behaviors, since experimental manipulations can introduce additional complexity (Revised Fig. 5 C-E). That said, we believe the strength of our approach lies in the large number of individual cells analyzed, which helps to mitigate variability and highlight consistent patterns. Furthermore, physiological heterogeneity likely exists within each organ as well, and by leveraging large-scale datasets, Immunemap can help reveal robust, population-level motility features that may otherwise be obscured.

Furthermore, we have analyzed the specific effect of the organs in the motility parameters of neutrophils during infection in both immune organs (lymph node and spleen) and in respiratory organs (lung and trachea). Interestingly, we have found significant differences in different parameters such as the speed (highest speed in respiratory organs compared to immune organs), or the arrest coefficient (higher in the lymph node and the trachea). We have included this analysis in the Revised Fig. 5 F-H. and in the Result and Discussion sections (Lines 281-307, and 400-409, respectively). As in the case of other types of analysis, we expect that the further addition of new data will strengthen the power of the comparison.

Figure 5. Immunemap enables large-scale comparative analysis of immune cell motility across diverse experimental conditions. (C) B cells under steady-state conditions (black) versus after antigen-specific stimulation (red). (D) OT-I CD8⁺ T cells at steady state (black) and following Poly I:C stimulation (red). (E) Neutrophils at early (black) and late (red) time points after injection of UV-inactivated PR8 virus. (F–H) Tissue-specific differences in neutrophil motility. Neutrophils exhibit increased speed in the airways compared to lymphoid tissues, highlighting how the tissue microenvironment influences migratory behavior.

- Detail: the authors often neglect the description of details necessary to understand major findings. E.g. the identification of DCs, which lack a lineage marker and need a combination of markers, it is unclear how these are identified (e.g. Fig 3).

We apologize for having omitted important details from the main text, it was not possible to include these informations in the manuscript for all the videos. These details along with staining and references to experimental protocols are made accessible through the Immunemap platform.

i.e. by opening the following video with DCs, the staining is specified. By clicking on “go to experiment”, or opening the second link, the details on the protocol and link to the main paper appears.

LINK1 staining: <https://app.immunemap.org/acquisition-public-view?id=187&videoID=250>

LINK2 exp details: <https://app.immunemap.org/experiment-public-view?id=49>

- Imaging modalities: a big issue in IVM is the use of different imaging platforms such as upright / inverted microscopes, 2-photon vs confocal, camera-based vs. laser scanning etc.

Could the authors provide more details how this affected their analysis and is there a way to use computational analysis to harmonize data obtained with different imaging tools?

We agree with the reviewer that the usage of different platforms can influence the cell migration measurements. Factors such as imaging volume, sampling rate, and photobleaching can all affect parameters like the duration of cell observation and the accuracy of speed estimation. In the revised version of the manuscript, we provide an example illustrating how time interval impacts measured speed (Revised Fig. 5 A-B). At this stage, a more comprehensive comparison between confocal and 2-photon microscopy is not feasible due to the unbalanced number of available videos. However, we anticipate that such analyses will become possible as the dataset grows.

Minor:

- Fig. 2J: The intention is to compare a video with loss of fluorescence (left) to a stable one (right). However, the right video has either loss of focus or bleaching as visible by the disappearance of green autofluorescence in the last sequence.

We thank the reviewer for the observation. Indeed, the intent of the video was to show loss of fluorescence due to an imaging artefact— specially, tissue drifting— which is now clearly stated in the figure legend. The image on the right (K) was included to contrast this example by showing a video with overall good quality parameters. As noted, some degree of imaging artifact is often unavoidable in IVM. In the case of panel J, there is a noticeable level of signal variation (SV) and photobleaching (PB), more apparent in the autofluorescence channel. While the combined quality score for this video is still relatively good, we acknowledge that some individual metrics reflect a lower performance compared to others.

The manuscript by Hickey et al curated a large database of single cell motility data derived from intravital imaging analysis in murine models. The team manually tracked and annotated over 58k single cell trajectories from 400 videos taken under different disease/treatment conditions. Based on these extensive single cell motility data, quantitative and unsupervised clustering analysis were performed to identify distinctive phenotypes of immune cell motility. The effort and care in curating this large, in vivo immune cell motility database are surely commendable, as the research community indeed needs such a database.

However, significantly more quantitative analysis of the annotated single cell tracks should be performed, especially considering that most of the single cell tracks are short, with only about 20 data points. Also, it will be very helpful and facilitate further design of intravital imaging experiments if the authors can analyze the data in a tissue and disease/treatment specific manner, instead of mostly lumping all the data together. The variations shown in Figure 3D-F are very large, indicating that it's not biologically meaningful to lump all the data together. Moreover, the textual discussions of the data analysis results should be substantially expanded so that the readers can know more about the biological insights derived from the data as well as the strengths and limitations of this intravital imaging dataset.

Major points:

1. Figure 3D-F: The standard deviations shown in these figures are very large for all immune cell types. To me, this indicates that the immune cells probably show different motility profiles, depending on the tissues and disease/treatment conditions. In Figure 3G, CD8 T cells clearly showed 2 distinctive clusters, suggesting two populations with very different motility characteristics. The authors should perform more detailed analysis of the immune cell motility specific to tissues and disease/treatment conditions. For example, it's interesting to even just know the immune cell motility in different tissues under the control condition (no disease, no treatment). Is T cell motility pretty much the same in lung and liver? What are the two populations of CD8+ T cells corresponding to? Significantly more data analysis are needed in order to truly understand the biological meaning of the data.

While we agree with the reviewer that the large variation does not allow us to perform comparisons of motility parameters by aggregating all the data together, we did not intend a comparative analysis (i.e., we did not perform a statistical test). Rather its main objective was to show how variable the motility patterns displayed by the imaged cells are. In this revised version we renamed the analysis as a "descriptive analysis" as we computed only mean and interquartile ranges, substituting the bar plots that might suggest a statistical analysis with violin plots serving as a 1d representation of the distinct motility patterns single cell types might display (Revised Fig. 3 D-F). In addition, we have analyzed the specific effect of the organs in the motility parameters of neutrophils during infection in both immune organs (lymph node and spleen) and in respiratory organs (lungs and trachea). Interestingly, we have found significant differences in different parameters such as the speed (highest speed in respiratory organs compared to immune organs), or the arrest coefficient (higher in the LN and the trachea). We have included this analysis in the Revised Figure 5 F-

H and in the result and Discussion sections (Lines 280-307, and 399-408, respectively). As in the case of other types of analysis, we expect that the further addition of new data will strengthen the power of the comparison.

2. A related question is how reproducible the intravital imaging measurements are. The authors curated data from 20 labs. It will be informative to know that data from these different labs are consistent and can indeed be grouped together to generate the database. A simple quality check is to compare single cell motility data of the same immune cell type under the same treatment condition, but acquired by different labs. For the 400 videos, are there such videos from different labs, i.e., with the same tissue model and treatment condition? If data from the different labs are consistent, it is reassuring for researchers that may want to use the database for further analysis. If the data from different labs are not consistent, it'll be helpful if the authors can identify possible origins of the experimental variability so that researchers can try to avoid such experimental variability when they design future intravital imaging experiments.

We agree with the reviewer that assessing the reproducibility and consistency of intravital imaging data across different laboratories is critical. In fact, addressing this variability was one of the primary motivations for creating Immunemap. The diversity in imaging protocols, tissue models, and experimental conditions across labs presents a known challenge. Indeed, we perform an initial quality validation regarding different technical parameters of the movies, setting up some quality standards (Fig 2 G-I). However, only a small subset of these datasets includes identical combinations of cell type, treatment, and tissue model across multiple labs, limiting the scope for direct reproducibility comparisons under strictly matched conditions. It is expected that this limitation will be miniced as the number of data increases. In this revised version we explain this limitation better in the discussion (Lines 373 - 383).

Moreover, we analyzed the effect of imaging parameters on motility metrics, finding that the temporal sampling rate had a strong impact on cells that are moving following non-linear paths, as detailed in the Revised Fig. 5 A-B. Volume size and tissue geometry also influenced some readouts, consistent with prior observations.

While cross-laboratory comparisons in similar conditions are not possible with the current amount of data included in Immunemap, this remains a key goal for future developments of the database.

Figure 5. Immunemap enables large-scale comparative analysis of immune cell motility across diverse experimental conditions. By providing access to a curated and standardized repository of intravital imaging data, Immunemap allows researchers to perform quantitative, cross-condition analyses of immune cell behavior. The platform facilitates the evaluation of how cell motility is influenced by factors such as temporal resolution, inflammatory stimuli, and tissue-specific environments—supporting data-driven discovery and hypothesis generation in immune imaging. **A–B**) Effect of temporal sampling rate on motility measurements. **(A)** Color-coded scatter plots showing the relationship between directionality and arrest coefficient across varying temporal resolutions. **(B)** Exponential decay in measured speed as a function of the time interval between consecutive image acquisitions. Data points represent 500 s track fragments from all Immunemap tracks with total durations >500 s.

3. Figure 4C: Unsupervised clustering is used to identify the unique clusters. In the Materials and Methods section, it's stated that 6 clusters were identified. But only 4 were shown in Figure 4C. What are the 2 clusters that are not shown? Can the authors perform principal component analysis (PCA) on the dataset and see whether the data would cluster differently when using, e.g., the 1st and 2nd PC to visualize the data?

We thank the reviewer for this helpful comment. In response, we have performed a new analysis using an updated clustering approach that yields more clearly defined cluster boundaries. Specifically, we generated the revised Fig. 4 using a refined pipeline applied to the datasets derived from 3D tracks. This clustering method, recently described by Dekkers et al. (2023), applies dynamic time warping (DTW) to compare entire cell tracks as time-series data, allowing for a more nuanced identification of motility patterns.

Following the reviewer's suggestion, we now provide a more detailed description of this clustering approach in the revised Methods section (Lines 490-501) and summarize the main findings in the Results section (Lines 228-278).

Regarding the discrepancy noted in the original version (six clusters mentioned, but only four shown in Fig. 4C), we appreciate the reviewer pointing this out. In the revised figure, all clusters are now included for clarity and completeness. Additionally, to further support the clustering results and provide an orthogonal visualization, we performed a dimensionality reduction via UMAP on the motility feature set and added a plot of the principal components to the revised figure. This analysis shows that the clusters identified through DTW-based

clustering are broadly recapitulated in the UMAP-reduced space, providing additional validation for the observed groupings.

We believe this updated analysis provides a more robust representation of the diversity of motility patterns and aligns well with the reviewer's request. We thank the reviewer again for helping us improve the clarity and rigor of this figure.

4. Have the authors calculated the Mean Squared Displacement (MSD) for the individual single cell tracks shown in Figure 4? A directionality index of 0.4-0.5 is not that large that would indicate non-random motion, while directionality index of 1 surely indicates directed motion. Grouping single cells with directionality of 0.4 with those close to 1 is not mechanistically meaningful to me. The authors should calculate and use more statistical motility parameters to characterize the single cell tracks (e.g., MSD) and perform the clustering analysis, especially considering the single cell tracks are mostly short.

We agree with the reviewer that the grouping presented in the previous version of the paper was not separating migration patterns indicative of random-walk and directed migration. This was due to an error in the uploading of the Figures presenting the results of K-Means instead of density peak clustering. We now updated the Figures. Moreover, we pre-processed tracks through DTW and dimensionality reduction, accounting also for mean square displacement as one additional feature. New results can be found in the Revised Fig. 4.

5. Figure 4E: Based on the clustering analysis, CD4 T cells seem to be the most motile with >50% showing low arrest coefficient and high directionality (i.e., persistent motion). This is different from the conclusion of CD8 T cells being most motile drawn from Figure 3D-F. Is this because the dataset used to derive Figure 4E is smaller than that used for Figure 3?

We agree with the reviewer on the highlighted difference. Indeed, the dataset was smaller. However, as outlined before, there was also a mistake in the uploaded Figure and reported clusters as outlined by discrepancies between the text and the Figure captions. In this revised version we reanalyzed the data, using 3D instead of 2D tracks and indeed we can confirm that CD8+ T cells displayed higher speed, mean square displacement, and directionality with respect to CD4+ T cells.

Again, it's important and informative to know the biological origins or correlation of the different motility phenotype distribution for the different immune cell types shown in Figure 4E. For instance, for the CD4 T cell tracks showing directed motion (denoted in green), are they mostly from the lymph nodes, while the arrest tracks are mostly from lung/liver (tissue-resident T cells)? The fact that the dataset is heavily biased towards lymph node data (Fig. 1D) makes such distribution analysis results not that meaningful because it's not really reflecting the motility phenotype distribution in vivo. It's more likely the result of data biased toward certain tissue and treatment condition. The authors should perform more careful analysis and take into account the highly heterogeneous numbers of data from the different tissues and treatment conditions (Fig. 1D) when presenting the data and drawing conclusions in Figure 4.

We agree with the reviewer that at the moment the amount of information included in the database does not allow extensive comparisons between the effects induced by a specific organ or condition. We expect that the addition of new data in the near future representative of different organs and conditions will strengthen this type of analysis. However, we have evaluated the specific effect of the organ in a collection of videos of neutrophils from different groups with comparable conditions. Following this analysis we have observed significant differences between the organs, as explained to reviewer 1, Revised Figure 5.

6. Figure 4F-H: This is very interesting and shows the biological insight that the database can provide. I would suggested the authors perform such analysis for all different immune cell types by comparing the control with the disease/treatment condition, and identify how the motility phenotypes may correspond to physiological or pathological processes. To me, this is one of the most impactful achievements of the database. It will also enable researchers to use the database for further quantitative analysis or as references. I understand this could involve a lot of figures. The authors can only show a few representative ones in the main text. The rest can be included as supplementary figures. The authors should also provide more discussions of the analysis results in the main text regarding the biological insight derived from the dataset.

We thank the reviewer for their thoughtful and encouraging comment. We fully agree that one of the most impactful aspects of Immunemap is its potential to uncover biologically meaningful patterns by comparing immune cell motility across conditions, such as control versus disease or treatment. As suggested, we have now included an example analysis in the revised manuscript where we compared neutrophil motility during infection across different tissues—specifically immune organs (lymph node and spleen) and respiratory organs (lung and trachea). This analysis revealed significant differences in several parameters, including cell speed (which was higher in respiratory organs) and arrest coefficient (higher in the lymph node and trachea), highlighting the biological insights that can be derived from the dataset (Revised Fig. 5F–H, Results section Lines 299-307, and Discussion section Lines 399-408).

We agree that extending this type of comparative analysis to other immune cell types would be highly valuable and aligns with the goals of Immunemap. However, due to current limitations in the number of annotated tracks for certain cell types and conditions, such an extensive analysis is not yet feasible across the full dataset. As the database continues to grow with new community contributions, we expect these comparisons to become increasingly comprehensive and informative. In the meantime, we believe that the current example serves as a strong proof of concept

References:

Pizzagalli, D., Farsakoglu, Y., Palomino-Segura, M. et al. *Leukocyte Tracking Database, a collection of immune cell tracks from intravital 2-photon microscopy videos*. *Sci Data* 5, 180129 (2018).

Dekkers, J.F., Alieva, M., Cleven, A. et al. *Uncovering the mode of action of engineered T cells in patient cancer organoids*. *Nat Biotechnol* 41, 60–69 (2023).

Dear Dr. Fernandez Gonzalez,

Thank you for the submission of your revised manuscript (EMBOJ-2025-120376R) to The EMBO Journal for our consideration, and for your patience during peer review. Your manuscript has been sent back to the three original referees who had previously assessed the first version of your manuscript, and we have now received their comments, which you can find below.

I am very pleased to say that all three referees are very satisfied with the revision and the strengthened manuscript, point out that this will be a valuable resource for the community, and support publication of the manuscript in The EMBO Journal. In light of this input, I would like to congratulate you on an excellent work and inform you that your manuscript has been in principle accepted for publication in our journal.

There are only 2 minor corrections needed -as identified by referee #2- which we need you to address in a final version of your manuscript.

In addition, and before we can move forward with publication of your manuscript, there are a few changes and corrections we need you to make in the final version:

- Please note that ORCID IDs are required from all co-corresponding authors; no ORCID ID has been linked yet to the profile of co-author Diego Ulisse Pizzagalli, and this must be completed before publication.

- Please remove subject categories, synopsis text and highlights, and synopsis image from the main manuscript file - see below instructions on how to upload this information.

- The members of the "The Immunemap project consortium" should be provided in the Appendix file instead of the main manuscript file.

- Please rename heading "Competing interests" to "Disclosure and competing interests statement".

- The author contributions statement should be removed from the manuscript file. Instead, we use CRediT to specify the contributions of each author in the journal submission system. Please feel free to use the free text box to provide more detailed descriptions during submission. See also our guide to authors for more information: <https://www.embopress.org/page/journal/14602075/authorguide#authorshipguidelines>.

- Please note that the format of the References is not correct and must be revised: it needs to be alphabetical rather than numerical, and "et al." needs to be used after the names of the first 10 co-authors of each publication. Please see our guide to authors for more information: <https://www.embopress.org/page/journal/14602075/authorguide#referencesformat>.

- We noticed that responses (pull-down menu for "Information included in the manuscript?") are missing in cells D87 and D88 of your Author Checklist; please revise and upload the corrected checklist.

- The provided funding information must be identical between the manuscript file ("Acknowledgements" section) and our online manuscript handling system; currently, the following information is missing from the online platform: "swissuniversities.ch via the Swiss Open Research Data Grant CHORD-B, a FIR grant from USI".

- "Supplementary" nomenclature, when referring to items included in the Appendix file, should be corrected throughout the manuscript (callouts) and the Appendix file: the correct nomenclature is "Appendix Figure S#", and "Appendix Table S#".

- The Appendix file needs a title page with heading "Appendix for:", followed by the manuscript's title, and a Table of Contents including page numbers for all listed items. The nomenclature (throughout the Appendix file and the callouts in the main manuscript file) should be "Appendix Figure S#".

- We will contact you soon with further instructions on how to best upload "Supplementary Table 1" and the ZIP folder "Supplementary Material 1" (which are currently uploaded as "Related manuscript files").

- Please note that EMBO press papers are accompanied online by:

- A) a short (2 sentences) summary of the findings and their significance,

- B) 2-5 short bullet points highlighting the key results, and

- C) a synopsis image in .jpg or .png format that is exactly 550 pixels wide and 300-600 pixels high (the height is variable). Please note that all text needs to be legible at the final size.

Please upload this information along with your revised manuscript (the text for A and B should be provided in a separate Word file).

- The materials and methods need to be described in the manuscript using our structured methods format, which is now required for all research articles. According to this format, the Methods section includes a single "Reagents and Tools Table" -listing key reagents, experimental models, software and relevant equipment including their sources and relevant identifiers- followed by a "Methods and Protocols" section describing the methods. Please download and fill our Reagents and Tools Table template (.docx), which you can find in our author guide:

<https://www.embopress.org/page/journal/14602075/authorguide#structuredmethods>. When submitting your revised manuscript, please do not include the Reagents and Tools Table in the Methods section of the manuscript but instead upload it as a separate file choosing the file type "Reagent Table".

- We were unable to locate the Source Data for Figure 4D - please explain in your Source Data checklist if these data are needed and provided.

- The Source Data need to be provided in a single ZIP folder per Figure, so that each folder (e.g. "Source Data Figure 1.zip", "Source Data Figure 2.zip" etc.) contains the source data for all panels of the respective Figure zipped together.

- During our routine data checks, our data editors have raised the following queries regarding figures and legends. Please make sure that the requests below are completely addressed in the final version of your manuscript (please highlight all changes in the revised manuscript):

- Please provide the exact p-values in the legend of Figure 5B.

- Please indicate the statistical test used for data analysis in the legend of Figure 5B.

- Please note that the box plots need to be defined in terms of minima, maxima, centre, bounds of box and whiskers, and percentile in the legends of Figures 2G, 5B, S2 C, E, G, I.

- Please note that information related to "n" is missing in the legends of Figures 2G, 3D-F; 5F-H; S2C, E, G, I; S4 D-F, G-I.

- Please note that scale bar and its definition are missing for Figure 1C.

- The order of the manuscript sections must be: Title page - Abstract and Keywords - Introduction - Results - Discussion - Methods - Data Availability - Acknowledgements - Disclosure and Competing Interests Statement - References - Figure Legends - main Tables (if there are any) - Expanded View Figure Legends.

Please also note that as part of the EMBO publications' Transparent Editorial Process, The EMBO Journal publishes online a Peer Review File along with each accepted manuscript. This File will be published in conjunction with your paper and will include the referee reports, your point-by-point response and all pertinent correspondence relating to the manuscript. You can opt out of this by letting the editorial office know (contact@embojournal.org). If you do opt out, the Peer Review File link will point to the following statement: "No Peer Review File is available with this article, as the authors have chosen not to make the review process public in this case."

We look forward to seeing a final version of your manuscript as soon as possible. Please let us know if you have any questions and use this link to submit your revision: <https://emboj.msubmit.net/cgi-bin/main.plex>.

Best regards,

Ioannis

Referee #1:

The authors have done an excellent job revising their submission in light of our previous comments. This revised paper will be of significant interest to many EMBO readers and provides an important example of the re-use of accumulated data in the imaging field.

They restored the 2D-based tracking to 3D tracking, using a FIJI plugin that allows "estimate z-coordinates" when manually tracking the cells in 2D.

They shortened the 500-second tracklets to 50 seconds for analysis, and used a dynamic time warping (DTW) based method to compare the tracklets. This substantially improved upon the initial 500-second duration tracks (where the more intricate changes in cell motility parameters would have been easily averaged out) and the DTW method allowed comparison against the other tracks rather than just using the "average values" to plot them on a 2D map and perform unsupervised clustering from them. This is a substantially improved analysis over the initially submitted ones.

They added new documentations and code examples to the computational pipelines that were missing in the initial submission.

Referee #2:

The study by Pizzagalli, Carrillo-Barbera et al. leverages over 400 intravital videos to create Immunemap, a curated dataset of immune relevant microscopy data, experiment-associated metadata, and immune cell tracks in living tissues. Immunemap is an important step to support reproducible imaging-based research within the community and ultimately infer novel dynamic characteristics of cells by combining multiple datasets across organs and conditions. In the revised version, the authors have satisfactorily address my concerns and I applaud the team for generating a fantastic resource for the community.

Minor:

- Fig. 1B page 3: instead of up / down it should be top / bottom.
- P7, ll 227-230: different font size

Referee #3:

In the revised manuscript, the authors provided substantial amount of new analysis results that showcase the usefulness and novelty of the Immunemap database, and they also updated the figures and manuscript text with more detailed description of the datasets, the analysis results and how the public can access and contribute to the database. The major issues that I raised for the original manuscript have largely been addressed satisfactorily. I understand that due to the limited size of the current database, it's impossible to perform some of the analysis that I suggested. I hope the authors will continue to grow and update this database and invite more researchers to contribute their annotated imaging data so that we will have a comprehensive database to mine and refer to for a wide spectrum of basic and translational research projects. Overall, I think the revised manuscript can be accepted for publication.

All editorial and formatting issues were resolved by the authors.

Dear Santiago,

Congratulations on an excellent manuscript! I am very pleased to inform you that it has been accepted for publication in The EMBO Journal. Thank you for comprehensively addressing the initially raised referee concerns and editorial requests for changes.

If you have any questions, please do not hesitate to contact the Editorial Office. Thank you for your contribution to The EMBO Journal. Working with you has been a pleasure.

Best regards,

Ioannis
